# The role of premature evidence accumulation in making difficult perceptual decisions under temporal uncertainty

Ciara A Devine[1]*, Christine Gaffney[1], Gerard M Loughnane[2], Simon P Kelly[3], Redmond G O'Connell[1]*

[1]Trinity College Institute of Neuroscience and School of Psychology, The University of Dublin, Trinity College, Dublin, Ireland; [2]National College of Ireland, Dublin, Ireland; [3]School of Electrical and Electronic Engineering and UCD Centre for Biomedical Engineering, University College Dublin, Dublin, Ireland

**Abstract** The computations and neural processes underpinning decision making have primarily been investigated using highly simplified tasks in which stimulus onsets cue observers to start accumulating choice-relevant information. Yet, in daily life we are rarely afforded the luxury of knowing precisely when choice-relevant information will appear. Here, we examined neural indices of decision formation while subjects discriminated subtle stimulus feature changes whose timing relative to stimulus onset ('foreperiod') was uncertain. Joint analysis of behavioural error patterns and neural decision signal dynamics indicated that subjects systematically began the accumulation process before any informative evidence was presented, and further, that accumulation onset timing varied systematically as a function of the foreperiod of the preceding trial. These results suggest that the brain can adjust to temporal uncertainty by strategically modulating accumulation onset timing according to statistical regularities in the temporal structure of the sensory environment with particular emphasis on recent experience.

**\*For correspondence:**
devineca@tcd.ie (CAD);
reoconne@tcd.ie (RGO'C)

**Competing interests:** The authors declare that no competing interests exist.

## Introduction

To date, perceptual decision making, described as the process whereby noisy sensory information is accumulated over time towards a decision bound, has been studied predominantly using tasks in which there is little ambiguity as to when observers should begin accumulating sensory information (*Gold and Shadlen, 2007*; *Ratcliff and Smith, 2004*; *Shadlen and Kiani, 2013*; *Smith and Ratcliff, 2004*). It has been documented extensively throughout the temporal attention literature, that there are significant advantages, across a wide-range of perceptual tasks, to knowing precisely when sensory evidence will be available (e.g. *Correa et al., 2006*; *Jepma et al., 2012*; *Niemi and Näätänen, 1981*; *Nobre et al., 2007*; *Nobre and van Ede, 2018*; *Vangkilde et al., 2012*). However, in the natural sensory world we are not always afforded the luxury of making decisions based on sensory events that have the same high degree of perceptual salience and/or predictability in terms of their onset timing as those typically used in experimental tasks. Such uncertainty presents a critical challenge because mistiming accumulation onset can have detrimental consequences: premature accumulation can lead to false alarms and fast errors while commencing accumulation late can lead to information loss and slow or missed responses (*Teichert et al., 2016*). The aim of this study was to shed light on the timing of evidence accumulation initiation when sensory evidence onsets are temporally uncertain and hard to detect.

Recent research has highlighted a few mechanisms that the brain could feasibly deploy. Several studies have pointed to the possibility that, under some circumstances, the brain relies on sensory-driven mechanisms that detect the appearance of goal-relevant sensory changes to trigger evidence accumulation. For instance, the Gated Accumulator Model, suggests that evidence accumulation onset is contingent on sensory input surpassing a threshold level (*Purcell et al., 2010*; *Purcell et al., 2012*; *Schall et al., 2011*). Similarly, evidence from human electrophysiology suggests that in a continuous monitoring context, where the timing of target sensory changes are unpredictable, early target selection mechanisms are involved in triggering evidence accumulation (*Loughnane et al., 2016*). However, the amplitude of target selection signals was found to be heavily dependent on the strength of the sensory evidence and therefore it is not clear how broadly such a stimulus-driven strategy can be applied. An additional possibility suggested by *Teichert et al. (2016)* is that the onset of sensory evidence accumulation may be subject to top-down control and adapted in accordance with the temporal context or demands of the sensory environment. Specifically, decision onset timing may be calibrated internally in order to optimise the trade-off between accumulating information too early, thereby risking performance errors, and accumulating too late, thereby losing valuable information. In the context of temporal uncertainty, behavioural modelling and psychophysics studies have indicated that prior knowledge of sensory evidence onset does modulate the timing of accumulation onset (*Bausenhart et al., 2010*; *Jepma et al., 2012*; *Seibold et al., 2011*), while others have pointed to |effects on the quality of sensory information (*Rohenkohl et al., 2012*). However, these studies have relied on indirect behavioural metrics for inferring accumulation onset time and direct neurophysiological evidence has been lacking.

In this study, we aimed to examine the mechanisms of perceptual decision making when subjects have imperfect foreknowledge of the precise timing of a subtle, goal-relevant sensory change. To this end, we conducted two experiments, both of which required subjects to discriminate a subtle change in a single stimulus feature (relative contrast of two overlaid gratings in experiment one and coherent dot motion in experiment 2). Temporal uncertainty was created by pseudorandomly imposing one of three delays intervening between stimulus onset and this sensory change (from hereon referred to as the 'foreperiod') on each trial. We traced activity in two previously characterised decision variable signals that trace sensory evidence accumulation: the centro-parietal positivity (CPP; *Kelly and O'Connell, 2013*; *Kelly and O'Connell, 2015*; *O'Connell et al., 2012*; *Twomey et al., 2015*) and effector-selective Mu/Beta band (10–30 Hz) activity (*de Lange et al., 2013*; *Donner et al., 2009*; *O'Connell et al., 2012*; *Siegel et al., 2011*; *Steinemann et al., 2018*). Each of these signals builds gradually during decision formation at a rate proportional to evidence strength, peaks around the time of a decision-reporting movement, predicts choice accuracy and RT across trials and exhibits amplitude modulations consistent with decision bound adjustments (*Kelly and O'Connell, 2013*; *O'Connell et al., 2012*; *Steinemann et al., 2018*; *Loughnane et al., 2016*; *Murphy et al., 2015*). More recently, the CPP has been further characterised within the same decision theoretic framework for a diverse range of perceptual tasks (*Afacan-Seref et al., 2018*; *Boubenec et al., 2017*; *Herding et al., 2019*; *Luyckx et al., 2019*; *Philiastides et al., 2014*; *Rungratsameetaweemana et al., 2018*; *Spitzer et al., 2017*; *von Lautz et al., 2019*; *van Vugt et al., 2019*), has been shown to correlate strongly with subjective reports of stimulus intensity (*Tagliabue et al., 2019*) and predict the timing and probability of choice error signaling (*Murphy et al., 2015*). Recent work has highlighted some key functional distinctions between the CPP and premotor Mu/Beta signals (*Twomey et al., 2016*). First, the evidence-dependent build-up of the CPP has been shown to reliably precede that of motor preparation signals (*Kelly and O'Connell, 2013*). Second, it has been shown across several studies that the CPP and motor signals undergo distinct strategic adjustments: premotor Mu/Beta-band activity contralateral to the decision reporting effector always reaches a stereotyped threshold level prior to response execution but both contralateral and ipsilateral signals exhibit systematic shifts in their starting levels in response to prior information about time constraints (*Steinemann et al., 2018*) and stimulus probability (*Kelly et al., 2019*), as well as a temporally increasing urgency component to their build-up (*Murphy et al., 2016*; *Steinemann et al., 2018*; *Kelly et al., 2019*). In contrast, the CPP has been found to not change its starting level and its pre-choice amplitude varied systematically as a function of RT for discrete decisions with a time limit (*Steinemann et al., 2018*; *Kelly et al., 2019*). Together, these data suggest that the CPP encodes a pure representation of cumulative evidence that is fed to the motor level where it is combined with other strategic influences.

We found that in this situation where subjects are unable to time the onset of decision formation to coincide with the uncertain onset of choice-relevant sensory evidence (contrast change in experiment one or coherent motion onset in experiment 2), they systematically initiated evidence accumulation in advance of the evidence. This was reflected in the build-up of the CPP during the foreperiod, which, in experiment 2, directly reflected random fluctuations in motion energy, and also in choice-predictive lateralisation of Mu/Beta band activity. The timing of accumulation onset relative to the sensory change accounted for the pronounced foreperiod effects that were observed on choice behaviour including increased miss rates on short foreperiod trials and elevated premature responses and fast errors on long foreperiod trials. Furthermore, we found that accumulation onset timing was not fixed across trials but varied systematically according to recent trial foreperiod duration, with accumulation commencing earlier if the previous foreperiod was shorter. This also gave rise to a congruency effect whereby choice accuracy was greater if the foreperiod duration on consecutive trials was congruent.

## Results

### Experiment 1: contrast discrimination

We analysed 128-channel EEG data from 23 human subjects performing a two-alternative contrast discrimination task (see Materials and methods; *Figure 1a*). in which they compared the contrast of two overlaid grating stimuli presented at fixation. At the outset of each trail, the gratings were presented and held constant at 50% contrast for one of three foreperiod durations (800 ms/1200 ms/ 1600 ms). Once the foreperiod had elapsed, the gratings underwent antithetical changes in contrast (one increased while the other decreased), and the participant was asked to report whether the left or right tilted grating was higher in contrast by clicking the corresponding mouse button with their corresponding thumb. The magnitude of the contrast change for each grating was determined separately for each subject during preliminary training using a staircase procedure (choice accuracy at 65–70%, mean contrast: 5.5% ± 1.35, see Materials and methods). Participants were informed clearly at the beginning of the study about the variability in the foreperiod durations. This was demonstrated to them during practise trials in which the contrast change was very large and evidence onset clearly noticeable. Points were awarded on each trial in order to encourage participants to respond both as quickly and as accurately as possible. 40 points were awarded for correct responses with a speed bonus of 0–40 additional points. Meanwhile, incorrect, missed or premature responses were awarded 0 points. Performance feedback was presented on a trial-by-trial (correct/incorrect/clicked too soon/too late) and block-by-block basis (accuracy, mean reaction time and cumulative points earned).

### Behaviour (experiment 1)

Taken together, our behavioural data suggest that subjects adopted a decision strategy that was more closely timed with respect to stimulus onset than to evidence onset. First, there were significant main effects of foreperiod on RT (*Figure 1b–c*; one-way repeated-measures ANOVA, $F_{(1.13, 22.56)}=1717$, $p=1.68 \times 10^{-39}$), accuracy (*Figure 1d–e*; $F_{(1.43, 28.64)}=60.08$, $8.91 \times 10^{-13}$), missed responses (*Figure 1f*; $F_{(1.07, 21.38)}=114.29$, $p=2.88 \times 10^{-17}$), premature responses (i.e. prior to sensory change onset, *Figure 1g*; $F_{(1.01, 20.12)}=39.55$, $p=1.38 \times 10^{-5}$) and points earned (*Figure 1h*; $F_{(1.17, 23.49)}=26.55$, $p=4.60 \times 10^{-8}$). Post-hoc tests (see *Figure 1b–h*; asterisks indicate significance levels) indicate that a longer (1600 ms) foreperiod resulted in faster but less accurate responses and an increased tendency to respond prematurely. The opposite was true of short (800 ms) foreperiod trials where responses were slower and less error prone but misses more common. Overall performance peaked on intermediate (1200 ms) foreperiod trials which yielded the highest levels of accuracy and greatest number of points earned per trial with relatively low levels of both misses and premature responses.

Next, we analysed accuracy as a function of RT separately for each foreperiod ('conditional accuracy functions'; *Figure 1d*). This analysis revealed two key trends. Firstly, the diminished accuracy on longer foreperiod trials manifested specifically on trials with faster RTs, reaching chance levels for the subset of long foreperiod trials with RTs closest to the contrast-change onset. This observation suggests that responses made close to the target change onset were based on little or no valid

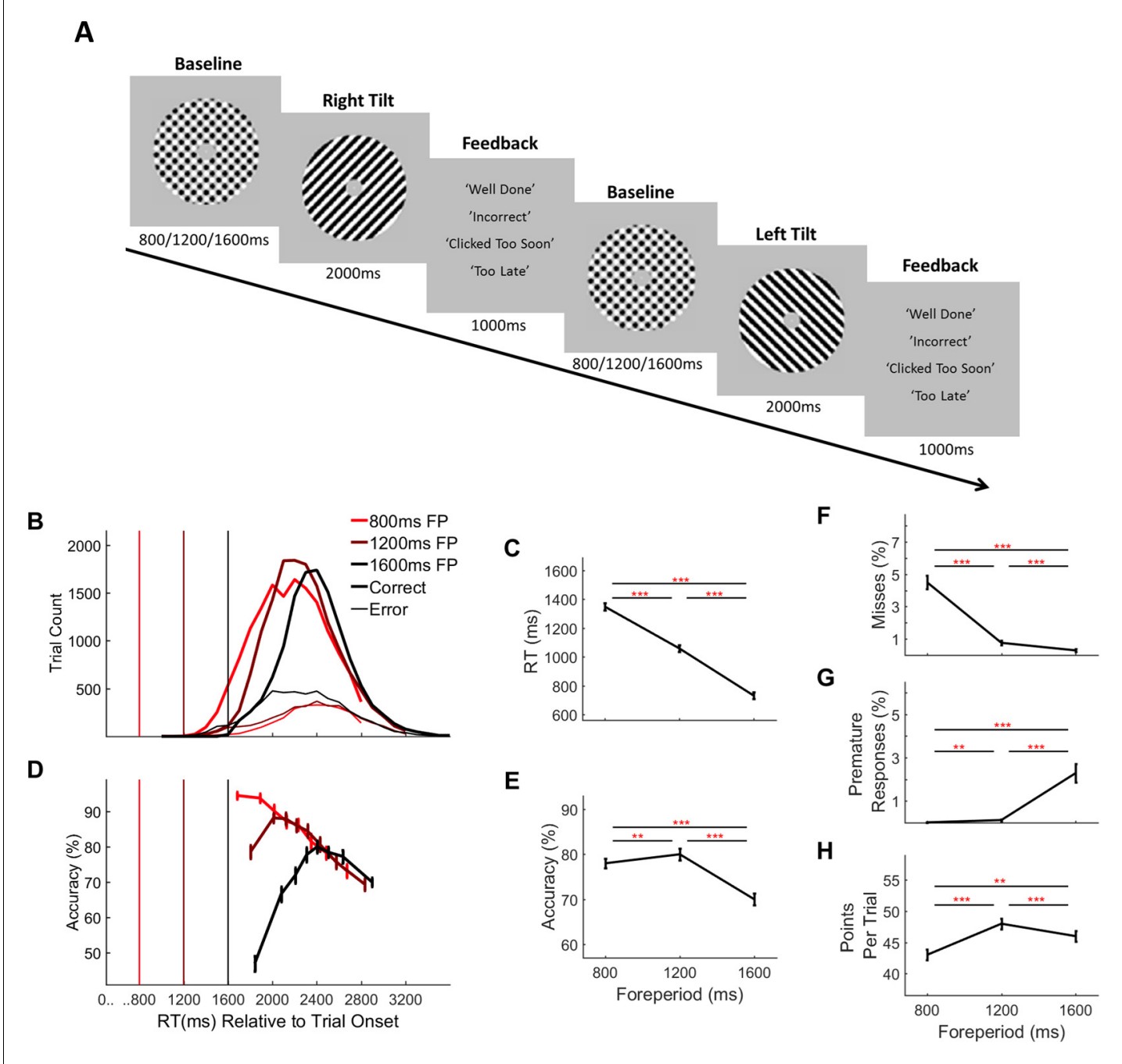

**Figure 1.** Two-alternative forced choice contrast discrimination task and behaviour separated by foreperiod (fp) duration. (**A**) Task Schematic: Each trial commenced with the presentation of two overlaid left- and right-tilted gratings (45° relative to the vertical midline) within a circular aperture against a grey background. At baseline the gratings were each presented at 50% contrast. After an initial foreperiod, the duration of which varied unpredictably from trial-to-trial (800 ms/1200 ms/1600 ms), one grating stepped up in contrast by an individually predetermined amount (*M* = 5.5%, *SD* = 1.35%, range = 2–7%) while the other stepped down by a corresponding amount. Schematic depicts a right-tilted target followed by a left-tilted target with 50% contrast changes for illustration purposes only. Feedback was presented at the end of each trial. (**B**) RT distributions separated by foreperiod and response accuracy. Vertical line markers indicate the times of the contrast change. (**C**) Mean RT (corrects and errors pooled) separated by foreperiod duration. (**D**) Conditional accuracy functions showing accuracy as a function of RT separated by foreperiod. Vertical line markers indicate the times of the contrast change. Mean accuracy (**E**), missed response rate (**F**), premature response rate (**G**) and points earned per trial (**H**), separated by foreperiod duration. Error bars represent standard error of the mean. Asterisks' indicate statistical significance of post-hoc comparisons: **=p < 0.017, ***=p < 0.001; Bonferroni corrected critical p-value=0.017.

The online version of this article includes the following figure supplement(s) for figure 1:

*Figure 1 continued on next page*

**Figure supplement 1.** Choice biases in contrast discrimination task.

sensory evidence. Secondly, the conditional accuracy functions also highlight that beyond 2400 ms, accuracy diminished monotonically as a function of RT across all three foreperiod durations.

## Sensory evidence representation unaffected by foreperiod duration

In order to determine whether sensory evidence representations were stable across the three foreperiod durations, we flickered each of the grating stimuli at a different frequency (20 Hz for left-tilted gratings, 25 Hz for right-tilted gratings) thereby eliciting separate SSVEPs in those corresponding frequency bands (20 Hz and 25 Hz respectively) over the occipital cortex (*Figure 2a*). Key characteristics of these SSVEP signals indicate that they provide a read-out of the representation of the sensory evidence for contrast-based perceptual decisions: they are highly sensitive to modulations of stimulus contrast (*Di Russo et al., 2007*; *Di Russo et al., 2001*; *Di Russo et al., 2003*; *Kim et al., 2007*; *Norcia et al., 2015*; *O'Connell et al., 2012*; *Vialatte et al., 2010*) and, their amplitudes predict both response accuracy (*Steinemann et al., 2018*) and RT (*Loughnane et al., 2018*; *O'Connell et al., 2012*).

We analysed the SSVEPs (pooling correct and error trials) as a function of foreperiod duration and stimulus (target vs non-target) using two-way repeated measures ANOVAs. As expected, the SNR of the target SSVEP (corresponding to the grating whose contrast increased) increased following the onset of the contrast change while that of the non-target (grating whose contrast decreased) decreased. Consistent with this we found a main effect of stimulus (target vs non-target) on SSVEP SNR in windows centred on 500 ms post-contrast change (*Figure 2b*; F(1, 18)=13.93, p=0.002) and 200 ms pre-response (*Figure 2c*; F(1, 18)=30.12, p=3.27×10$^{-5}$). These effects are also apparent in the difference SSVEP (d-SSVEP), which was calculated by subtracting the non-target from the target SSVEP (*Figure 2d–e*). Foreperiod on the other hand did not affect the SSVEP SNR (Post-contrast

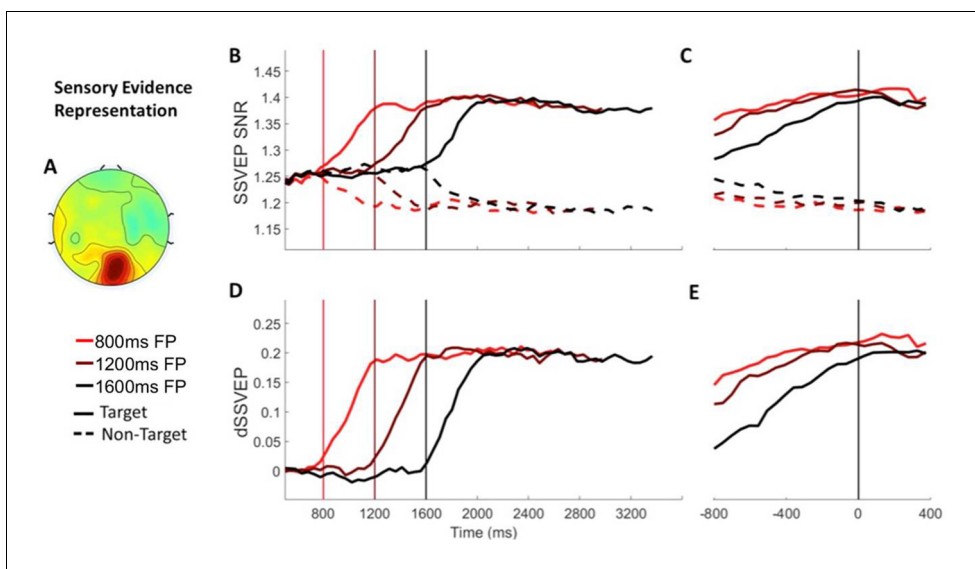

**Figure 2.** Sensory evidence representation (SSVEP) Signals separated by foreperiod (FP). (**A**) The topography of the d-SSVEP measured prior to response was maximal over the occipital cortex (Oz). (**B**) Stimulus-aligned target and non-target SSVEP signals separated according to foreperiod duration, plotted relative to the onset of the stimulus. Vertical line markers at 800/1200/1600 ms indicate the times of the contrast change. (**C**) Response-aligned target and non-target SSVEP signals separated according to foreperiod duration. The vertical line marker at 0 ms denotes the time of response. (**D**) Stimulus-aligned d-SSVEP signal separated according to foreperiod duration, plotted relative to the onset of the stimulus. Vertical line markers at 800/1200/1600 ms indicate the times of the contrast change. (**E**) Response-aligned d-SSVEP signal separated according to foreperiod duration. The vertical line marker at 0 ms denotes the time of response.

change: F(2, 36)=0.06, p=0.94; Pre-response: F(1 18.40)=0.01 p=0.92), irrespective of the stimulus (foreperiod x stimulus interactions: Post-contrast change: F(2, 36)=0.45, p=0.64; Pre-response: F (1.05, 18.96)=0.13, p=0.73). Together these findings indicate that the representation of sensory evidence was stable across the three foreperiods and could not account for the observed behavioural trends.

## Premature decision formation on trials with longer foreperiods

*Figure 3b* shows that, on longer foreperiod trials, significant CPP build-up was observed prior to the onset of the contrast change (*Figure 3b*; one-sample t-tests: 800 ms: t(18)=1.79, p=0.14; 1200 ms: t (18)=3.71 p=0.0016; 1600 ms t(18)=5.52, p=$2\times10^{-5}$; correct and error responses pooled). Considering that these data were collected over a period of five consecutive experimental sessions we sought to establish whether this early build-up of the CPP during the foreperiod might be a phenomenon that emerges only after extensive practice. In fact, there was significant pre-evidence build-up of the CPP on longer foreperiod trials as early as session one (*Figure 3—figure supplement 1*; t(18)=4.12, p=6.4×10-5) and the amount of pre-evidence CPP-build up did not vary systematically across training sessions (main effect of session: F(4, 72)=1.18, p=0.33).

To explore this early CPP buildup further, we analysed the CPP as a function of foreperiod duration and RT using two-way repeated measures ANOVAs. To this end we separated the data, within each foreperiod, into six equally sized bins, according to RT. Our analyses found that the CPP amplitude prior to evidence onset was larger on longer foreperiod trials (F(1.27, 22.91 = 25.00, p=$1.6\times10^{-5}$; linear contrast: p=$4.75\times10^{-5}$; quadratic contrast: p=0.17). Moreover, there was an inverse relationship between the amount of pre-evidence CPP build up and RT (*Figure 3d*; F(5, 90) =6.56, p=$3\times10^{-5}$), irrespective of foreperiod duration (RT x foreperiod interaction: F(5.33, 95.91) =1.07, p=0.39), indicating that greater pre-evidence CPP build-up was associated with faster responses. Further, the build-up rate of the CPP during the foreperiod (measured as the slope of a line fit to the CPP waveform between −250 ms and 50 ms relative to the contrast change) also predicted RT, irrespective of foreperiod duration (*Figure 3f*; main effect of RT: F(5, 90)=4.42, p=0.001; main effect of foreperiod: F(2,36)=2.39, p=0.11; RT x foreperiod interaction: F(4.81 86.54)=0.63, p=0.67).

In fact, an overall inverse relationship between pre-evidence accumulation and RT should not necessarily be expected: whereas pre-evidence accumulation should accelerate responses when it happens to favour the ultimately chosen alternative, it would slow responses when it happens to favour the ultimately unchosen alternative because it would serve to push the decision process further from its final bound. Thus, the degree to which a statistically significant inverse relationship between pre-evidence CPP and RT would manifest would depend on the prevalence of this latter subset of 'change of mind' trials. Such changes of mind should be far more prevalent on correct trials and accordingly, we observed a significant RT by Choice Accuracy (RT x Choice (correct vs incorrect) interaction: F(3, 54)=3.94, p=0.02; *Figure 3—figure supplement 2b*) reflecting a stronger relationship between pre-evidence CPP and RT on error trials.

Further interrogation of the CPP-RT relationship observed in experiment one revealed that pre-evidence accumulation was systematically biased in favour of left choices. Subjects were 4.17 times more likely to choose left when responding prematurely (<150 ms; t(18)=3.20, p=0.005; *Figure 1— figure supplement 1a*), 1.95 times more likely to respond left when responding quickly (<500 ms; t (20)=5.62, p=0.1.69$^{-4}$; see also *Figure 1—figure supplement 1b*) and had faster RTs when left targets were presented (F(1, 20)=63.65, p=1.22$^{-7}$; *Figure 1—figure supplement 1c*). This systematic bias can be attributed to the difference in flicker frequency between left (20 Hz) and right-tilted (25 Hz) gratings. Psychophysical research has previously shown that within this range the perceived brightness (*Bartley, 1938*; *Bartley, 1951*; *Wu et al., 1996*) and contrast (*Solomon and Tyler, 2018*) of flickering stimuli diminishes as a function of flicker frequency, even when they are matched in terms of physical contrast and luminance. Since the left-tilted grating appeared higher-contrast during the foreperiod despite physical equivalence, this may have caused pre-evidence accumulation to be biased toward left choices resulting in fewer changes of mind when the left choice was the correct one. Indeed pre-evidence CPP declined significantly more steeply as a function of RT for left choices than it did for right choices (RT x Choice interaction: F(3, 54)=3.19, p=0.03; *Figure 3—figure supplement 2a*). Thus, it is likely that the overall statistically significant relationship between pre-

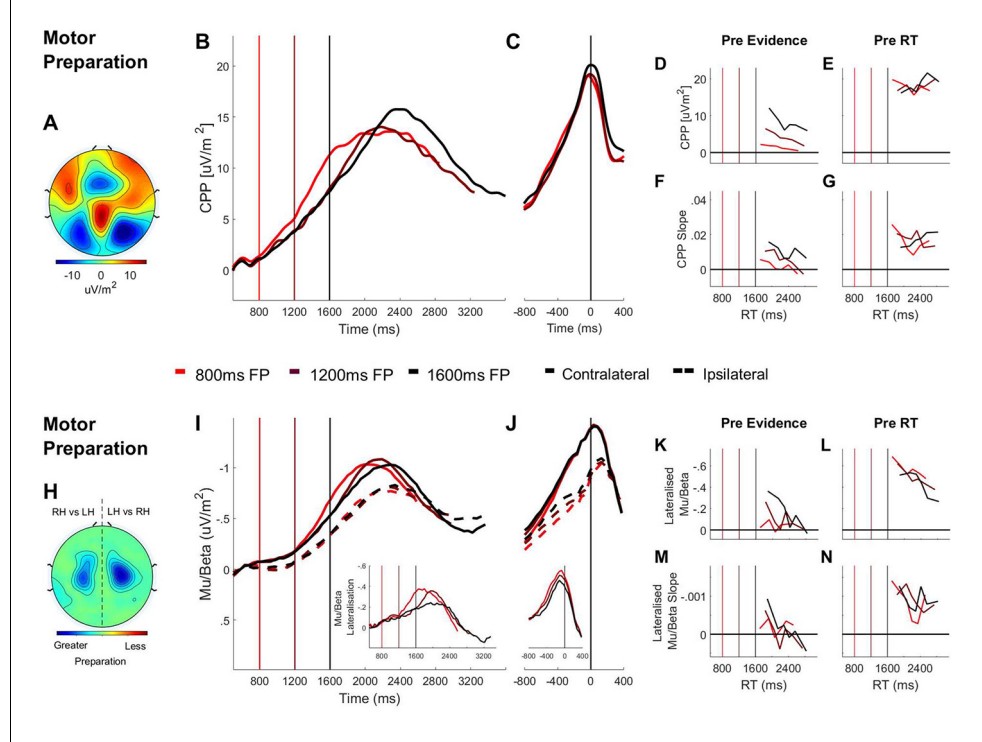

**Figure 3.** Domain-general (CPP) and effector-selective (Mu/Beta 10–30 Hz). Decision Signals Separated by Foreperiod (FP) in the Contrast Discrimination Task. (**A**) Topography of the ERP signal measured prior to response (−150 ms to −50 ms) showing a positive going centroparietal component maximal over Pz. (**B**) Stimulus-aligned CPP separated by foreperiod duration, plotted relative to the onset of the overlaid gratings stimulus. Vertical line markers at 800/1200/1600 ms indicate the times of the contrast change across the three levels of foreperiod duration. (**C**) Response-aligned CPP separated by foreperiod duration. The vertical line marker at 0 ms denotes the time of response. (**D**) CPP amplitude measured at contrast change (−50 ms to 50 ms) and (**E**) at response (−150 ms to −50 ms) plotted as a function of RT separately for each foreperiod. (**F**) Pre-evidence CPP build-up rate (−250 ms to 50 ms) and (**G**) pre-response CPP build-up rate (−500 ms to −200 ms), plotted as a function of RT separately for each foreperiod. (**H**) Topography of lateralised Mu/Beta band (10–30 Hz) activity measured prior to response (−150 ms to −50 ms) calculated separately for each hemisphere by subtracting ipsilateral from contralateral hand responses (LH = left hand; RH = right hand).The topography shows stronger lateralisation over each hemisphere when preparing a contralateral response. (**I**) Stimulus-aligned contralateral and ipsilateral Mu/Beta waveforms, separated by foreperiod duration, plotted relative to the onset of the overlaid gratings. Vertical line markers at 800/1200/1600 ms denote the times of the contrast change across the three levels of foreperiod duration. Insert: stimulus-aligned Mu/Beta lateralisation (contralateral-ipsilateral) traces. (**J**) Response-aligned contralateral and ipsilateral Mu/Beta waveforms, separated by foreperiod duration with a vertical line marker at 0 ms denoting the time of response. Insert: response-aligned Mu/Beta lateralisation (contralateral-ipsilateral) traces. (**K**) Mu/beta lateralisation at contrast change (−50 to 50 ms) and (**L**) response (−150 ms to −50 ms), plotted as a function of RT separately for each foreperiod. (**M**) Pre-evidence Mu/Beta lateralisation slope (−250 ms to 50 ms) and (**N**) pre-response Mu/Beta lateralisation slope (−500 ms to −200 ms) plotted as a function of RT separately for each foreperiod.

The online version of this article includes the following figure supplement(s) for figure 3:

**Figure supplement 1.** Pre-evidence stimulus-aligned CPP waveforms on long (1600ms).

**Figure supplement 2.** Pre-evidence cpp amplitude on long foreperiod (1600ms).

**Figure supplement 3.** Contralateral and ipsilateral motor preparation as a function of pre-evidence CPP amplitude bin (large versus small based on median split).

**Figure supplement 4.** Bilateral Occipital ERP aligned to Contrast Change.

evidence CPP and RT in this experiment arises from this choice bias which reduced the prevalence of change of mind trials.

We also measured and analyzed CPP build-up rates in the response-aligned waveforms. There was no main effect of either RT (F(5, 90)=0.75, p=0.59) or foreperiod duration (F(2, 36)=0.56, p=0.58) but there was a significant cross-over interaction (*Figure 3g*; F(10, 180)=4.24, p=2.7×10$^{-5}$), suggesting that the relationship between RT and pre-response CPP build-up rate was reversed for short relative to long foreperiod trials. *Figure 3g* shows that when the foreperiod was short, faster RTs were associated with steeper pre-response CPP slope, in line with previous observations (e.g. *Kelly and O'Connell, 2013*; *O'Connell et al., 2012*). By comparison, the opposite was true of long foreperiod trials where faster responses were associated with shallower pre-response CPP slope. The shallower build-up rate of the pre-response CPP on faster RT trials can be explained in light of the close temporal proximity of these responses to the timing of the contrast change and hence the lesser contribution of valid sensory evidence to those decisions.

Alongside the early CPP build-up, we observed a progressive increase in motor preparation reflected in Mu/Beta desynchronisation over the hemispheres contralateral and ipsilateral to the response-executing hand throughout the pre-evidence foreperiod (*Figure 3i*). To specifically isolate the choice-predictive component of motor preparation, we examined the differential motor preparation for the ultimately executed response relative to the withheld response, by computing the Mu/Beta lateralisation index (contralateral minus ipsilateral to the response hand). This analysis indicated that there was significant pre-evidence Mu/Beta lateralisation on longer foreperiod trials (*Figure 3i* inset; one-sample t-tests: 800 ms: t(18)=-1.27, p=0.22; 1200 ms: t(18)=-3.64, p=0.002; 1600 ms: t(18) =-3.91 p=0.001). To further examine this choice-predictive motor activity we analysed Mu/Beta lateralisation as a function of foreperiod and RT using two-way repeated measures ANOVAs. Similar to the CPP, longer foreperiods gave rise to greater pre-evidence Mu/Beta lateralisation (F(2, 36)=3.64, p=0.04) which in turn scaled inversely with RT, irrespective of foreperiod duration (*Figure 3k*; main effect of RT: F(5, 90)=2.94, p=0.017; RT x foreperiod interaction: F(5.22, 93.99)=1.68, p=0.14). Thus greater choice-selective motor preparation during the pre-evidence foreperiod predicted faster RTs. In addition, the rate of Mu/Beta lateralisation during the foreperiod (also measured as the slope of a line fit to the Mu/Beta lateralisation waveform between −250 ms and 50 relative to evidence onset) predicted RT irrespective of foreperiod duration (*Figure 3m*; main effect of RT: F(5, 90)=3.56, p=0.006; main effect of foreperiod: F(2, 36)=0.47, p=0.63; RT x foreperiod interaction: F(4.82, 86.70) =1.45, p=0.22). Also in line with the findings reported for the CPP, there was no main effect of foreperiod on the pre-response build-up rate (measured between −450 and −150 ms) of Mu/Beta lateralisation (F(2, 36)=0.25, p=0.78) but there was a main effect of RT (F(5, 90)=5.13, p=0.01) which was dependent on foreperiod duration (foreperiod x RT interaction: F(10, 180)=2.08, p=0.03). This interaction indicates that the predicted pattern of faster pre-response Mu/Beta lateralisation build-up rates for earlier responses occurred only on short foreperiod trials but not when the foreperiod was longer (*Figure 3n*). To further examine possible interactions between biased motor preparation in the pre-evidence window and the early build-up of the CPP we divided the long foreperiod data of each subject into two bins based on a median split of the single-trial pre-evidence CPP amplitude values (excluding trials with RT <0 and>500 ms relative to evidence onset) and tested for cross-bin differences in motor preparation prior to the CPP measurement window (*Figure 3—figure supplement 3*). No such differences were observed for the window 800–1600 ms (t(18)=1.12, p=0.28; all p>0.1 for t-tests on contra-ipsi lateralisation in contiguous 50 ms bins from 800 ms to 1600 ms). This observation accords with recent demonstrations that prior-infomed motor level adjustments do not impact on the CPP process (*Steinemann et al., 2018*; *Kelly et al., 2019*).

## Subtle sensory evidence onsets do not elicit early target selection responses

Recent research and models of perceptual decision making have pointed to the importance of sensory-driven target selection mechanisms in triggering the onset of evidence accumulation (*Purcell et al., 2010*; *Purcell et al., 2012*; *Schall et al., 2011*). In the human brain, this target selection role has long been attributed to a negative potential over occipital scalp elicited approximately 200–300 ms after the onset of a goal-relevant sensory stimulus (*Luck, 2012*; *Luck and Hillyard, 1994*). Recently we showed that the target selection process is invoked bilaterally even for a single

stimulus at fixation that has no ostensible need for 'selection' and that in a continuous monitoring context, where target onsets are completely unpredictable, the signal temporally precedes and has a predictive influence on the onset and rate of evidence accumulation (*Loughnane et al., 2016*). This highlights the potential general role for this target selection signal in triggering the accumulation process under conditions of temporal uncertainty, but given its reliance on the presence of strong, detectable stimulus transitions (*Loughnane et al., 2016*), it is not clear whether it could play a major role in the current task. In fact, here, we did not observe any deflection in bilateral occipital ERP waveforms in the N2 time window although there was a slow, sustained negativity over the bilateral occipital cortex (*Figure 3—figure supplement 4a–b*). Thus, in contrast to previous continuous tasks with larger evidence steps, low-level target-selection processes do not appear to play a role in dealing with temporal uncertainty in the present context.

## Foreperiod duration does not affect decision signal amplitudes at response

Recent neurophysiological studies have highlighted that when decisions must be made to a strict deadline, time-dependent urgency signals serve to expedite the decision process, progressively lowering the quantity of cumulative evidence required to trigger commitment (*Steinemann et al., 2018*; *Murphy et al., 2016*; *Hanks et al., 2014*; *Thura and Cisek, 2016*). In a recent study we showed that these urgency effects manifested in a time-dependent reduction in CPP amplitude (*Steinemann et al., 2018*). Here, however, we did not find any variation in the CPP amplitude prior to response as a function of foreperiod duration ($F_{(2, 36)}=0.47, p=0.63$; *Figure 3c*) or response time ($F_{(5, 90)}=0.64$, p=0.67; *Figure 3e*). Similarly, the amplitude of contralateral Mu/Beta at the time of response did not vary as a function of foreperiod duration (*Figure 3j*; $F_{(2, 36)}=0.49$, p=0.63) or response time (*Figure 3n*; $F_{(2.16, 38.95)}=1.80$, p=0.18), consistent with a time-invariant action-triggering threshold at the motor level (*Murphy et al., 2016*; *Steinemann et al., 2018*). By comparison, ipsilateral Mu/Beta prior to response, though not modulated significantly by foreperiod duration ($F_{(1.46, 26.30)}=2.82$, p=0.09), was more desynchronised prior to response on slower RT trials ($F_{(1.86, 33.40)}=6.50$, $p=3\times10^{-5}$) reflecting more preparation of the unchosen response hand on these trials and a closer 'race' between the alternative outcomes. Correspondingly, mu-beta lateralisation prior to response was stronger on faster response trials ($F_{(2.68, 48.31)}=3.99$, p=0.01), a finding that is consistent with previous studies (*Murphy et al., 2016*).

## Summary: experiment 1

The key results from experiment one show that in the absence of foreknowledge about the precise onset timing of a subtle goal relevant sensory change, subjects were unable to align the onset of decision formation to sensory evidence onset. On longer foreperiod trials, subjects consistently commenced decision formation before the target contrast change occurred, resulting in a higher incidence of false alarms and fast errors. The premature onset of the decision process was reflected in the build-up of the CPP and the emergence of choice predictive Mu/Beta lateralisation during the foreperiod. However, these data do not allow us to definitively establish whether the early build-up of decision-related activity reflected the accumulation of sensory noise, as opposed to some other process such as urgency (e.g. *Cisek et al., 2009*; *Thura et al., 2012*; *Thura and Cisek, 2016*). To address this in experiment 2, we used the random dot motion task in which variability in physical dot motion energy from one frame to the next can be directly quantified by applying dot motion energy filtering. Random variations in motion energy during the foreperiod were leveraged in order to directly test for behavioural and electrophysiological indications of premature evidence accumulation.

## Experiment 2: dot motion discrimination

We analysed 128-channel EEG data from 23 human subjects who performed a two-alternative forced choice random dot motion discrimination task in which participants were required to judge the dominant direction of motion in a cloud of moving dots (see Materials and methods; *Figure 4a*). On each trial the dots initially moved randomly (0% coherence) for the duration of an initial foreperiod that varied unpredictably from one trial to the next across three levels (800 ms/1200 ms/1600 ms). Once the foreperiod had elapsed, a portion of the dots began to move coherently in either a leftward or rightward direction. The portion of coherently moving dots, which was determined separately for

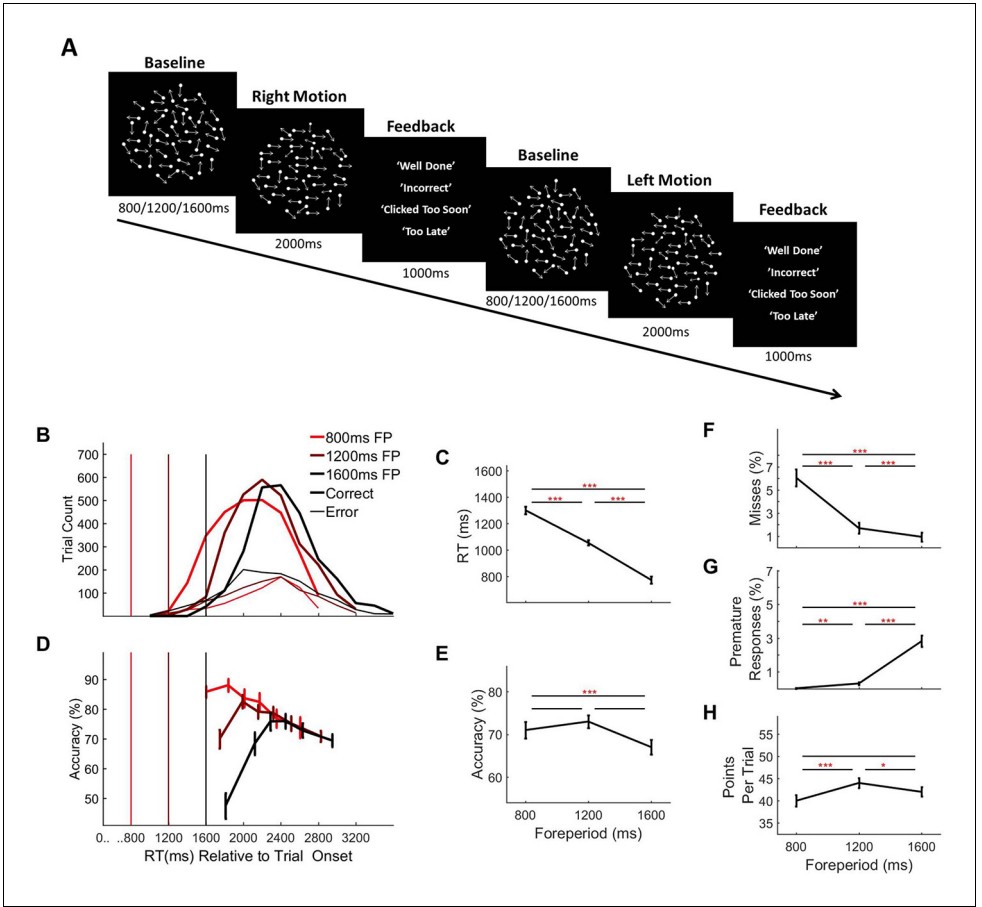

**Figure 4.** Random dot motion discrimination task and behaviour separated by forepriod (fp) duration. (A) Task Schematic: Each trial commenced with the presentation of a cloud of moving dots within a circular aperture against a black background. At baseline, the dots were displaced at random from one trial to the next. After an initial foreperiod, the duration of which varied unpredictably from trial-to-trial (800 ms/1200 ms/1600 ms), a portion of the dots, determined separately for each individual subject ($M$ = 7.42%, $SD$ = 2.61%, range = 3–12%), began to move coherently in either a leftward or rightward direction. The schematic depicts a rightward motion trial followed by a leftward motion trial (Note: Arrows are used here to illustrate the direction of motion but were not a feature of the actual stimulus). Feedback was presented at the end of each trial. (B) RT distributions separated by foreperiod duration and response accuracy. Vertical line markers indicate the time of coherent motion onset. (C) Mean RT separated by foreperiod duration and (D) Conditional accuracy functions showing accuracy as a function of RT separated by foreperiod duration. Vertical line markers indicate the time of coherent motion onset. Mean accuracy (E), missed response rate (F), premature response rate (G) and points earned per trial (H), separated by foreperiod. Error bars represent standard error of the mean. Asterisks' indicate statistical significance of post-hoc comparisons: *=<0.05, **=p < 0.017, ***=p < 0.001; Bonferroni corrected critical p-value=0.017.

The online version of this article includes the following figure supplement(s) for figure 4:

**Figure supplement 1.** Choice biases in contrast discrimination task.

each subject using a staircase procedure (see Materials and methods) was, on average, 7.42% ($SD$ = 2.61%). Subjects were required to indicate the perceived direction of the coherent motion by clicking the corresponding (left/right) mouse button with their corresponding thumb (left/right). Participants were fully aware of the initial foreperiod and received feedback and points based on their performance following the same method implemented in experiment 1.

## Behaviour

As in experiment 1, we first analysed behaviour separated according to foreperiod duration (summarised in *Figure 4b–h*) and observed highly similar trends including significant effects on RT

(*Figure 4b–c*; F(1.07, 19.20)=150.60, p=1×10⁻¹⁰), accuracy (*Figure 4d–e*; F(1.47, 26.38)=7.69, p=0.005), miss rate (*Figure 4f*; F(1.08, 19.42)=63.76, p=9.7×10⁻⁸), premature response rate (*Figure 4g*; F(1.08, 19.40)=58.80, p=1.8×10⁻⁷) and points earned per trial (*Figure 4h*; F(142, 25.50) =10.10, p=0.002).

## Decision formation commenced prematurely on longer foreperiod trials and was predictive of behaviour

The results from experiment two broadly replicate those of experiment 1. As before, there was significant build-up of the CPP prior to the onset of the coherent motion (*Figure 5b*) for longer foreperiod trials (one-sample t-tests: 800 ms: t(17)=1.21 p=0.24; 1200 ms: t(17)=2.28, p=0.03; 1600 ms: t(17)=3.49, p=0.003), with longer pre-evidence foreperiods resulting in larger CPP amplitudes at the time of coherent motion onset (F(2, 34)=5.45, p=0.009). Also in line with experiment 1, there was no effect of foreperiod duration on the build-up rate of the CPP either pre-evidence (F(1.45, 24.57) =1.96, p=0.17) or pre-response (F(2, 34)=0.04, p=0.96). Similarly, we observed desynchronisation in Mu/Beta band activity over the hemispheres contralateral and ipsilateral to the response-executing hand throughout the pre-evidence foreperiod, with a progressive lateralisation toward the contralateral hemisphere on longer foreperiod trials (*Figure 5i*; 800 ms t(17)=-.40, p=0.69; 1200 ms: t(17)=-1.65, p=0.12; 1600 ms: t(17)=-3.23, p=0.005; main effect of foreperiod: F(3, 34)=3.49, p=0.04). The rate of Mu/Beta lateralisation did not vary across foreperiod durations either during the pre-evidence period (F(3, 34)=0.72, p=0.49) or pre response (F(2, 34)=0.09, p=0.92). Like experiment one the CPP amplitude prior to response did not vary significantly as a function of foreperiod duration (*Figure 5c*; F(1.29, 21.92)=0.04, p=0.96) or response time (*Figure 5e*; F(2, 34)=0.52, p=0.6). Likewise, the amplitude of contralateral Mu/Beta at the time of response did not vary as a function of foreperiod duration (*Figure 5j*; main effect foreperiod: F(2, 34)=0.13, p=0.88) or response time (*Figure 5j*; main effect response time: F(2, 34)=0.74, p=0.48).

Although the principle findings of experiment one were replicated in experiment 2, confirming the early onset of the decision formation process on longer foreperiod trials, some trends from experiment one were notably absent in experiment 2. Specifically, the relationship between RT and pre-evidence CPP amplitude (F(2, 34)=0.12, p=0.89), pre-evidence CPP slope (F(1.45, 24.57)=1.96, p=0.16) and pre-response CPP slope (F(2, 34)=2.08, p=0.14) did not reach statistical significance. Likewise, there was no relationship between RT and Mu/Beta lateralisation in the pre-evidence (Amplitude: F(2, 34)=1.73, p=0.19; Slope: F(2, 34)=0.71 p=0.50) or pre response time periods (Amplitude: F(2, 34)=1.80, p=0.18; Slope: F(2, 34)=0.33, p=0.72). We believe that the absence of these effects can be attributed to the fact that, unlike in experiment 1, there was no evidence that premature responses were biased in favour of either choice outcome (*Figure 4—figure supplement 1a–b*; Ratio of Left to Right choices = 0.88; t(16)=-.69, p=0.50; subjects with fewer than 10 premature choices were excluded) and no difference in RT between left and right targets (*Figure 4—figure supplement 1c*; Left Responses: *M* = 998 ms, *SD* = 89.28 ms; Right Responses: *M* = 1091 ms, *SD* = 117.37 ms; F(1, 21)=0.94, p=0.34). This is unsurprising given that the movement of the dots during the foreperiod was entirely random and thus, by comparison with experiment one where left and right stimuli differed in flicker frequency, it was not possible for the sensory evidence to systematically favour either choice. In the absence of any bias early accumulation would have the effect of speeding up responses on some trials (e.g. when evidence during the foreperiod by chance favoured the upcoming target or was strong enough to cause subjects to cross a decision bound) and slowing responses on others (change of mind trials). It is also possible that experiment two was less sensitive to RT effects compared to experiment one due to the lower trial numbers that were acquired per subject.

## Pre-evidence cumulative motion energy predicts fast choices and pre-evidence CPP build-up

In order to quantify the degree to which the dot motion stimulus by chance favoured one decision alternative or the other during the incoherent motion foreperiod, we performed dot motion energy filtering (see Materials and methods; *Adelson and Bergen, 1985*; see also *Urai et al., 2017*). On the basis that the effects of random motion energy fluctuations should be most prominent for early choices, we separated long foreperiod trials into fast (<500 ms) and slow responses (>500 ms) and

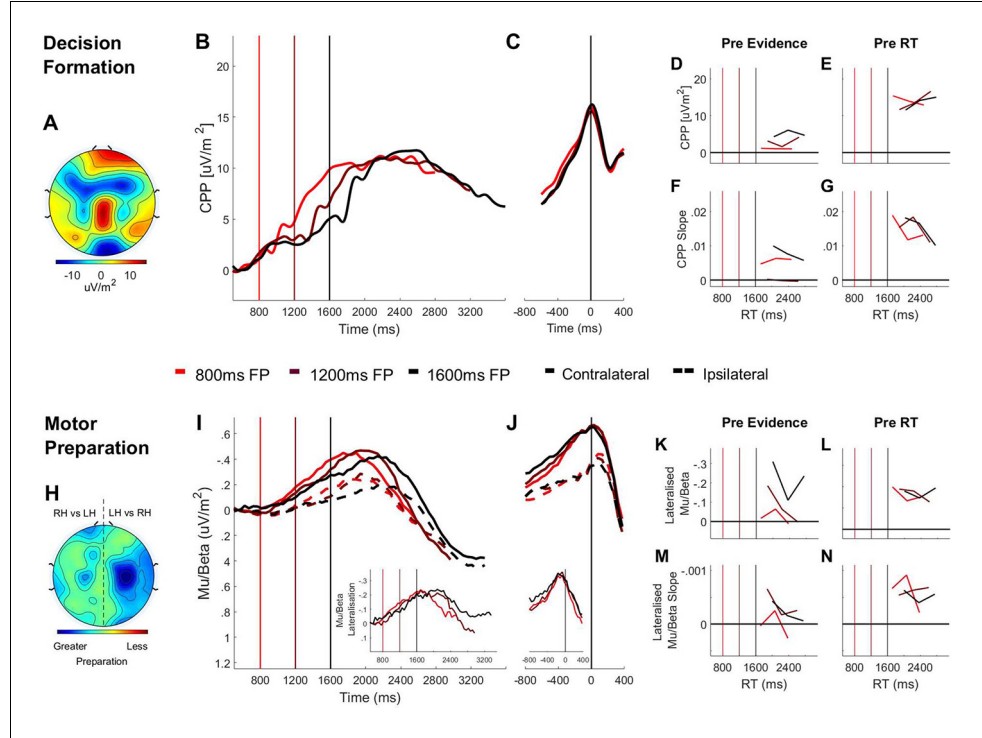

**Figure 5.** Domain-general (CPP) and effector-selective (Mu/Beta 10–30 Hz). Decision signals separated by foreperiod (FP) in the random dot motion discrimination task. (**A**) Topography of the ERP signal measured prior to response (−150 ms to −50 ms) showing a positive going centroparietal component maximal over Pz. (**B**) Stimulus-aligned CPP separated by foreperiod duration, plotted relative to the onset of the dot motion stimulus. Vertical line markers at 800/1200/1600 ms indicate the times of coherent motion onset across the three levels of foreperiod duration. (**C**) Response-aligned CPP separated according to foreperiod duration. The vertical line marker at 0 ms denotes the time of response. (**D**) CPP amplitude measured at coherent motion onset (−50 ms to 50 ms) and E) at response (−150 ms to −50 ms) plotted as a function of RT separately for each foreperiod. (**F**) Pre-evidence CPP build-up rate (−250 ms to 50 ms) and G) pre-response CPP build-up rate (−500 ms to −200 ms), plotted as a function of RT separately for each foreperiod. (**H**) Topography of lateralised Mu/Beta band (10–30 Hz) activity measured prior to response (−150 ms to −50 ms) calculated separately for each hemisphere by subtracting ipsilateral from contralateral hand responses (LH = left hand; RH = right hand). (**I**) Stimulus-aligned contralateral and ipsilateral Mu/Beta waveforms separated by foreperiod duration, plotted relative to the onset of the dot motion stimulus. Vertical line markers at 800/1200/1600 ms denote the times of coherent motion onset across the three level of foreperiod duration. Insert: stimulus-aligned Mu/Beta lateralisation (contralateral-ipsilateral) traces. (**J**) Response-aligned contralateral and ipsilateral Mu/Beta waveforms, separated by foreperiod duration with a vertical line marker at 0 ms denoting the time of response. Insert: response-aligned Mu/Beta lateralisation (contralateral-ipsilateral) traces. (**K**) Mu/beta lateralisation at coherent motion onset (−50 to 50 ms) and L) response (−150 ms to −50 ms), plotted as a function of RT separately for each foreperiod. (**M**) Pre-evidence Mu/Beta lateralisation slope (−250 ms to 50 ms) and N) pre-response Mu/Beta lateralisation slope (−500 ms to −200 ms) plotted as a function of RT separately for each foreperiod.

The online version of this article includes the following figure supplement(s) for figure 5:

**Figure supplement 1.** Bilateral occipital erp aligned to coherent motion.

---

then calculated the integral of dot motion energy during the foreperiod (100 ms-1600ms). One-sample t-tests confirmed that there was significant sustained motion energy in the direction of the chosen response on fast RT trials (*Figure 6a*; t(17)=2.21 p=0.04) but not on slow RT trials (*Figure 6a*; t(17)=.03, p=0.98).

Next we sought to determine whether pre-evidence motion energy predicted the magnitude of pre-evidence CPP build-up. To this end we separated trials into two bins according to the slope of the pre-evidence dot motion integral (shallow vs. steep). This enabled us to to isolate trials where the cumulative motion energy during the foreperiod strongly and consistently favoured one choice

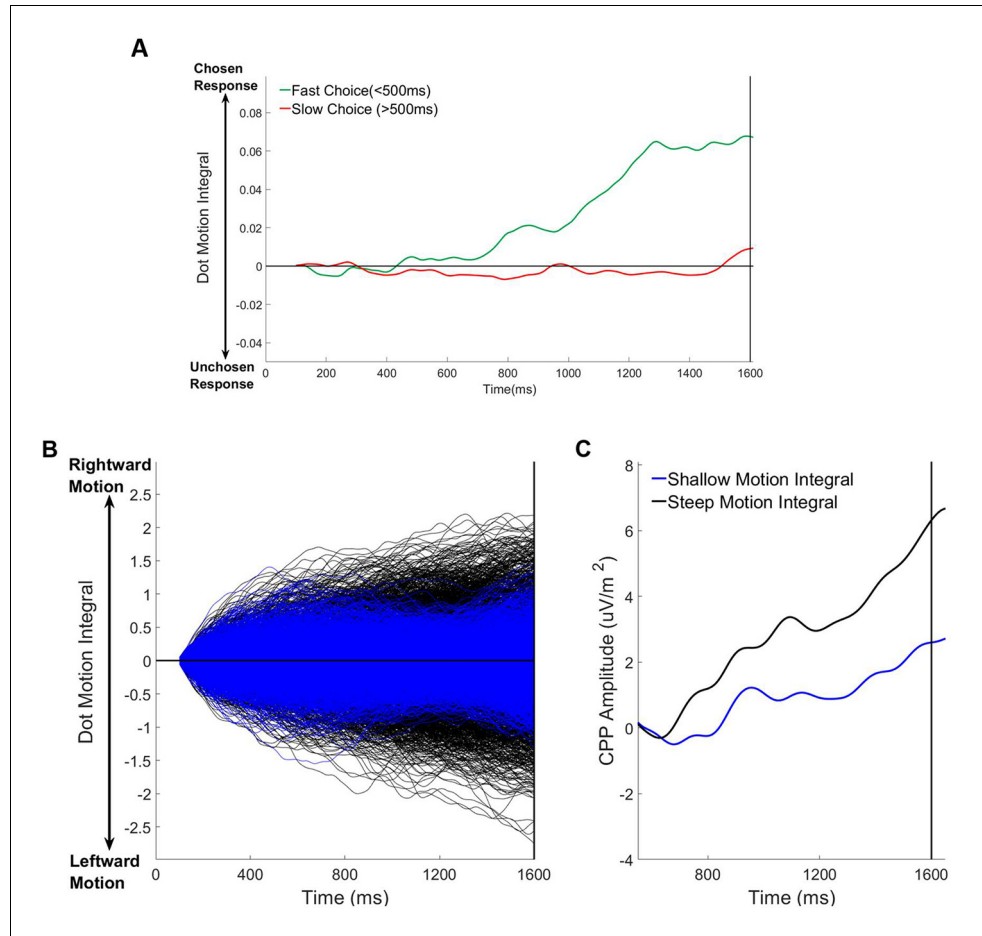

**Figure 6.** Relationship between pre-evidence cumulative dot motion energy, choice and pre-evidence cpp build-up (1600ms foreperiod trials only). (**A**) Average cumulative dot motion integral (measured between 100–1600 ms) separated for fast (RT <500 ms) and slow (RT >500 ms) responses. (**B**) Single trial cumulative dot motion integral traces separated according to the slope of the pre-evidence dot motion integral (100–1600 ms). (**C**) Grand average stimulus-aligned CPP separated according to the slope of the the pre-evidence dot motion integral.

or another by chance alone (*Figure 6b*). For each subset of trials we measured the slope of the pre-evidence CPP over a broad window (600 ms-1600ms) and the amplitude of the CPP in a 100 ms window centred on the time of coherent motion onset (1600 ms). We found that there was significant pre-evidence build-up of the CPP on trials where the cumulative motion energy consistently favoured a particular choice (*Figure 6c*; one-sample t-tests: Amplitude: t(17)=3.21 p=0.005; Slope: t(17)=3.25, p=0.005) but not on those where the dot motion weakly and/or inconsistently favoured a specific choice (*Figure 6c*; Amplitude vs 0: t(17)=1.37, p=0.19; Slope vs 0: t(17)=1.59, p=0.13). Paired samples t-tests comparing this pre-evidence CPP build-up between the two conditions (shallow vs steep pre-evidence motion integral) were significant (Amplitude: t(17)=2.41 p=0.03) or close-to-significant (Slope: t(17)=1.92, p=0.07) suggesting that the build-up of the CPP did reflect cumulative motion energy during the foreperiod.

## Subtle sensory evidence onsets do not elicit early target selection responses

As in experiment 1, we were interested in establishing the extent to which target selection mechanisms played a role in decision formation in the context of discriminating a subtle change in a sensory stimulus. Here, we observed a small, N2-like deflection in the time window 200–300 ms relative to coherent motion onset over the bilateral occipital regions (*Figure 5—figure supplement 1a–b*).

However, relative to baseline, the magnitude of this component was not statistically significant for any of the foreperiods (all p>0.5 for t-tests against 0). Again, this suggests that low-level target-selection processes are not reliably elicited when coherent motion is weak.

## Decision onset timing is modulated by recent temporal structure

Thus far our data suggest that foreperiod effects on choice performance on these tasks can be attributed, in large part, to the premature onset of evidence accumulation occurring in this task. Next, we sought to examine the degree to which evidence accumulation onsets varied systematically as a function of trial history. Prior research in humans (*Jazayeri and Shadlen, 2010*) and in monkeys (*Jazayeri and Shadlen, 2015*) suggests that we exploit previous experiences and statistical knowledge in order to overcome uncertainty when estimating time or anticipating events. To further explore this idea we conducted additional analyses of the data from experiment 1, hypothesising that, if decision onset timing is indeed affected by prior knowledge of temporal structure, then recent experiences may be particularly influential. For instance, in the temporal attention literature, the widely-documented sequential foreperiod effect suggests that the speed with which perceptual decisions are made is affected by previous trial timings (*Capizzi et al., 2013*; *Los and Heslenfeld, 2005*; *Mento, 2013*; *Mento, 2017*; *Van der Lubbe et al., 2004*), though this phenomenon has not explicitly been linked or investigated with respect to the timing of evidence accumulation onset. Here, we separated trials according to the foreperiod on the current trial (foreperiod$_n$) and the foreperiod on the immediately preceding trial (foreperiod$_{n-1}$). We found that behavioural performance on foreperiod$_n$ was modulated significantly by foreperiod$_{n-1}$, such that responses were faster when foreperiod$_{n-1}$ was shorter (*Figure 7a*; $F(2, 40)=140.17$, $p=2.49\times10^{-19}$) and this effect was more pronounced when foreperiod$_n$ was long (foreperiod$_n$ x foreperiod$_{n-1}$ interaction: $F(4, 80)=7.37$, $p=4.17\times10^{-5}$). Moreover, foreperiod$_{n-1}$ also affected accuracy ($F(2, 40)=5.23$, $p=0.01$), but this effect was dependent on foreperiod$_n$ (foreperiod$_n$ x foreperiod$_{n-1}$ interaction: $F(4, 80)=14.90$, $p=3.85\times10^{-9}$). The trends shown in *Figure 7b* indicate that on short or long foreperiod$_n$ trials, foreperiod repetition led to a gain in accuracy whereas a change in foreperiod duration from one trial to the next was associated with a cost to accuracy. Furthermore, in line with the above effects, we found that the rate of missed responses on short foreperiod$_n$ trials was reduced if short foreperiods occurred consecutively (*Figure 7c*; $F(2, 40)=36.51$, $p=9.52\times10^{-10}$) while the rate of premature responses on long foreperiod$_n$ trials was reduced when long foreperiods occurred consecutively (*Figure 7d*; $F(2, 40)=8.12$, $p=0.001$).

Our analyses further show that the effects of foreperiod$_{n-1}$ on behaviour can be attributed, at least in part, to trial-by-trial modulation of evidence accumulation onset timing. As illustrated in *Figure 8a*, the CPP reached a higher amplitude at the time of the contrast change when foreperiod$_{n-1}$ was shorter ($F(2, 36)=4.38$, $p=0.02$). The absence of any effect of foreperiod$_{n-1}$ on pre-evidence CPP slope ($F(2, 36)=0.54$, $p=0.59$) suggests that the pre-evidence amplitude differences arose from differences in the onset, rather than the rate, of premature evidence accumulation. Together

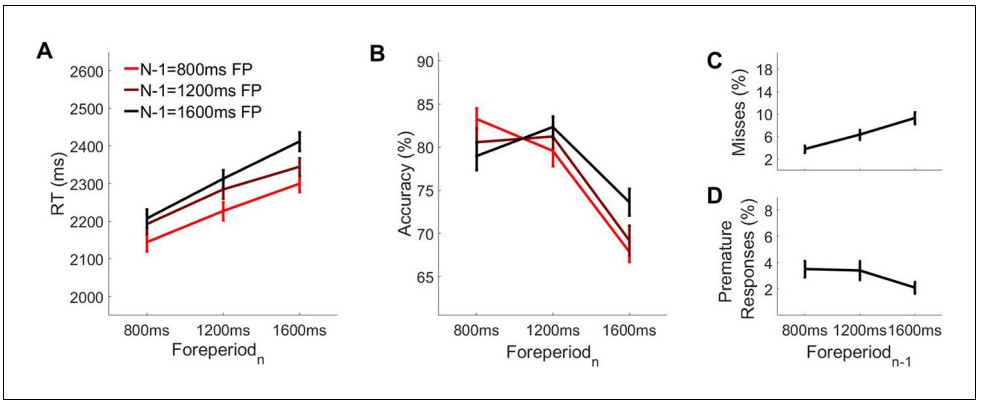

**Figure 7.** Behavioural performance as a function of foreperiod$_{n-1}$ and foreperiod$_n$. Mean accuracy (A) and mean RT (B) as a function of foreperiod$_{n-1}$ and foreperiod$_n$. (C) Missed response rate on short foreperiod rials as a function of foreperiod$_{n-1}$was long. (D) Premature response rate on long foreperiod trials as a function of foreperiod$_{n-1}$.

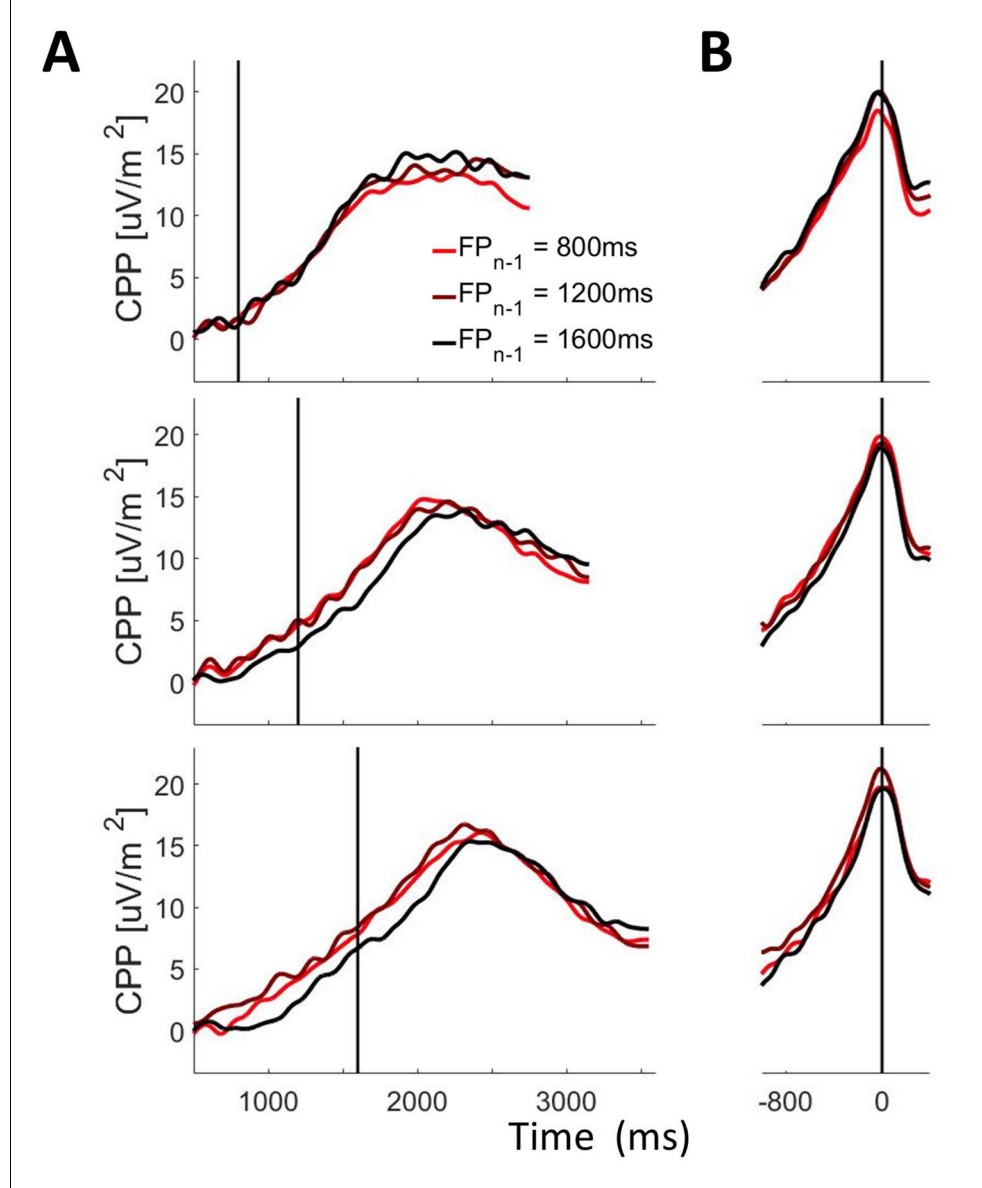

**Figure 8.** CPP as a function of foreperiod$_n$ and foreperiod$_{n-1}$. (A) Stimulus-aligned and (B) Response-aligned CPP waveforms separated according to foreperiod$_n$ (Top Panel: 800 ms; Middle Panel 1200 ms; Bottom Panel 1600ms) and foreperiod$_{n-1}$. Shaded grey bars correspond to the pre-evidence (A) and pre-response (B) measurement windows. Solid black vertical lines mark the time of evidence onset (A) and response (B).

these findings indicate that decision onset timing is modulated by recent experience of temporal structure in the sensory environment.

## Discussion

In this study we examined the effects of temporal uncertainty on evidence accumulation onset timing. Convergent behavioural and neurophysiological data reveal that when evidence onsets are difficult to discern, participants initiate the accumulation of sensory information endogenously, before the actual evidence appears.

Recent work has highlighted that when sensory evidence onsets are unpredictable but relatively salient, early target selection signals are involved in triggering evidence accumulation onset (*Loughnane et al., 2016*; see also *Purcell et al., 2010*; *Purcell et al., 2012*; *Schall et al., 2011*).

Here, our titration procedure ensured that the onset of the sensory changes were very weak and, by consequence, sensory driven target selection signals were not reliably observed in our data, forcing participants to rely on alternative strategies. Importantly, and in line with the observations in *Teichert et al. (2016)*, our data indicate that participants adapted the onset timing of accumulation strategically, using their knowledge and recent experience of the underlying temporal structure of the task. For instance, the CPP data suggest that subjects in this study timed the onset of evidence accumulation to approximately coincide with the earliest possible time point at which evidence could be presented to them (800 ms). Rather than optimising overall performance on shorter foreperiod trials (which had a notably high rate of missed responses and earned the lowest number of points), this strategy instead led to a peak in performance on intermediate foreperiod trials (which had notably high levels of accuracy and low levels of both missed and premature responses thus earning a higher number of points). One speculative explanation for this is that subjects may have adopted a Bayesian approach, 'regression to the mean', in estimating the response deadline, (*Jazayeri and Shadlen, 2010*; *Jazayeri and Shadlen, 2015*). Given that the foreperiod durations used in this experiment were equally probable, this strategy would cause subjects to bias their timing estimates of the impending deadline towards the trial duration on intermediate foreperiod trials. This is an interesting avenue for further investigation and could be examined by repeating this study, drawing the foreperiod duration from various asymmetrical distributions. More generally, the above highlights that several features of the present paradigm, including subtlety of evidence onsets, time pressure and temporal uncertainty were key drivers of the observed premature evidence accumulation. An important challenge for future work will therefore be to establish the degree to which the present findings generalise to other paradigms and to precisely delineate the task parameters that encourage/discourage pre-evidence accumulation.

Although the overall temporal structure of the task clearly affected evidence accumulation, our data indicate that subjects did not simply time the onset of evidence accumulation to a fixed point on all trials. Here we show that the timing of evidence accumulation onset is, at least in part, calibrated based on the duration of the foreperiod on recent trials, with shorter foreperiods leading to earlier accumulation onset on subsequent trials and correspondingly faster RTs. Moreover, trial-by-trial adjustments in accumulation onset timing were also consequential for accuracy. When the same foreperiod duration was, by chance, repeated from one trial to the next, decision onset timing was more aligned to the physical sensory change than when the foreperiods were mismatched, and this incurred a corresponding benefit for accuracy. In turn, when consecutive foreperiods were highly mismatched (i.e. short followed by long or vice-versa), there was greater misalignment of sensory evidence onset relative to the physical sensory change and, correspondingly, greater decrements in accuracy were incurred. According to the normative theories, intertrial dependencies such as this may reflect the inherent tendency for the brain to monitor and exploit local statistical patterns in the environment in order to reduce uncertainty (*Jones et al., 2013*; *Wilder et al., 2013*; *Wilder et al., 2010*; *Yu and Cohen, 2008*). Thus, these trial-by-trial adjustments in accumulation onset timing may reflect continuous updating of temporal expectations that inform accumulation onset timing.

With regard to the precise nature of the evidence accumulated during the foreperiod on later onset trials, our data show that this, at least partly, reflects the accumulation of sensory noise. In experiment two, we were able to show that early CPP build-up was driven by random fluctuations in motion energy, which, on some trials, cumulatively favoured one choice or another. However, we do not exclude the possibility that the CPP may be subject to other influences. For instance, many studies have shown that individuals exhibit a variety of idiosyncratic choice-history biases that manifest as trial-by-trial adjustments in the starting point of evidence accumulation (*Bode et al., 2012*; *de Lange et al., 2013*; *Yu and Cohen, 2008*; *Zhang et al., 2014*) or biases in the weighting of sensory evidence (*Braun et al., 2018*; *Kloosterman et al., 2018*; *Talluri et al., 2018*; *Urai et al., 2019*). It remains to be established whether such biases may be directly represented in the CPP. Nevertheless, our findings add to and complement the ever-growing body of research into serial dependency in perceptual decision making and speak directly to the question of how temporal uncertainty affected decision onset timing.

Our results also have the potential to shed light on the mechanisms underpinning a well-known phenomenon in the temporal attention literature, known as the sequential foreperiod effect. This effect, which is another example of serial dependency, is characterised by the modulation of RT by previous trial foreperiod in the context of target detection tasks with variable foreperiods

(*Capizzi et al., 2013*; *Capizzi et al., 2012*; *Capizzi et al., 2015*; *Los et al., 2017*; *Los and Van Den Heuvel, 2001*; *Steinborn and Langner, 2012*; *Vallesi et al., 2009*; *Vallesi et al., 2007*). Our results replicate the sequential foreperiod effect using a task in which respondents performed a difficult perceptual discrimination task where the foreperiod varied pseudorandomly from trial to trial. In this context at least, the sequential foreperiod effect, the underlying mechanisms of which have not yet been fully explained, can be attributed to the adjustment in evidence accumulation onset timing. Interestingly, in the broader literature on temporal orienting, data from psychophysical and computational modelling studies have implicated decision onset timing in temporal orienting more generally (*Bausenhart et al., 2010*; *Jepma et al., 2012*; *Rolke, 2008*; *Rolke and Hofmann, 2007*; *Seibold et al., 2011*). Our findings provide the first neurophysiological evidence in support of this idea.

Finally, our study also highlights how neural decision signals can be leveraged to gain insights into the timing of accumulation onset that are otherwise very difficult to discern through behaviour. Previous studies investigating decision onset timing have relied on indirect behavioural measures such as fast error rate, shifts in the leading edge of response time distributions, and estimates of non-decision time derived from sequential sampling models (*Teichert et al., 2016*). The use of non-decision time is problematic because the timing of evidence accumulation is just one of a number of factors, alongside afferent (e.g. sensory encoding, neural transmission) and efferent (e.g. motor execution) decision-related processes, that contribute to this parameter. Our data highlight that it is possible to infer the timing of accumulation onset directly from the temporal dynamics of the CPP.

In summary, this study has shed light on the important role of accumulation onset timing in perceptual decision-making and has yielded two key insights in this regard. Firstly, we have shown that in the absence of foreknowledge about the precise timing of a subtle goal relevant sensory change, observers are unable to accurately time the onset of sensory evidence accumulation with respect to that event, resulting in instances where accumulation commences too early, leading to premature responses and fast errors, as well as instances where accumulation is delayed, leading to missed responses. Secondly, we have demonstrated that accumulation onset timing is not governed solely by sensory driven mechanisms but instead can be controlled by top-down processes. To this end, we appear to rely on and continuously update our average representation of the temporal structure of a given task.

## Materials and methods

### Subjects

Experiment 1: contrast discrimination task

Twenty-three subjects aged 19–30 (12 males, 11 females) were recruited to take part in a two-alternative, forced choice, contrast discrimination task as part of a study on perceptual learning taking place over the course of 6 separate experimental sessions. This pre-planned sample size is consistent with previous electrophysiological studies conducted within our group (e.g. *Murphy et al., 2015*; *O'Connell et al., 2012*; *Kelly and O'Connell, 2015*; *Steinemann et al., 2018*; *Twomey et al., 2015*) and elsewhere (e.g. *Diaz et al., 2017*; *McGovern et al., 2012*; *McGovern et al., 2012*). All subjects were right-handed, had normal or corrected-to-normal vision and no history of personal or familial neurological or psychiatric illness. Subjects were requested to abstain from alcohol consumption and to maintain regular patterns of sleeping for the duration of the study so as to minimise potential confounds between sessions. One subject was excluded after reporting retrospectively that they had failed to comply with this prerequisite. One further subject dropped out after the second session and their data were excluded. Data from the final experimental session were excluded for one additional subject due to an error in setting the presentation monitor refresh rate on that day. This resulted in a final sample size of 21 (10 male, 11 female; Age: $M = 22.41$, $SD = 2.96$). For the purpose of electrophysiological analyses two further subjects were excluded, due to excessive blink and/or EEG artefacts (>40% trial loss), but their data were retained for behavioural analyses.

Prior to taking part, subjects were informed that they would receive a €10 reimbursement per testing session with the possibility of earning a performance bonus of up to €20 upon completing the study. In order to incentivise subjects, they were misled to believe that the exact value of their bonus was dependent on how many points (see design and procedure) they accumulated over the

course of the study. In fact, all subjects who completed the study were awarded the maximum additional bonus of €20 (total payment = €80) after being debriefed about the nature of the study and rationale for the deception. Written, informed consent was obtained from all subjects and all procedures were approved by the Trinity College Dublin ethics committee and conducted in accordance with the Declaration of Helsinki.

## Experiment 2: dot motion discrimination task

Nineteen subjects aged 18–37 (*M* = 25.74, *SD* = 5.92; 10 females, nine males) were recruited to take part in a dot motion discrimination task comprising a single session of data collection. This pre-planned sample size is consistent with previous electrophysiological studies conducted within our group (e.g. *Murphy et al., 2015*; *O'Connell et al., 2012*; *Kelly and O'Connell, 2015*; *Steinemann et al., 2018*; *Twomey et al., 2015*). All subjects were right-handed, with normal or corrected-to-normal vision and had no history of personal or familial neurological or psychiatric illness. In the case of one subject, a file containing the data necessary for carrying out analyses of dot motion energy (see below) was missing. Their data were retained for all behavioural and electrophysiological analyses with the exception of those involving dot motion energy data.

Subjects were informed that they would earn €15 for taking part in the study with the possibility of earned an additional performance bonus of up to €10. In order to incentivise subjects, they were misled to believe that the exact value of the bonus was dependent on how many points they obtained over the course of the study. In fact all subjects were awarded the maximum bonus for completing the study after being debriefed about the nature of the study and rationale for the deception. Written, informed consent was obtained from all subjects and all procedures were approved by the Trinity College Dublin ethics committee and conducted in accordance with the Declaration of Helsinki.

## Experiment 1: two-alternative forced-choice contrast discrimination task

Subjects performed a difficult two-alternative contrast discrimination task which was programmed in Matlab (Mathworks, Natick, MA, USA), using the Psychtoolbox-2 (http://psychtoolbox.org/) package. The task code is available at (https://github.com/CiaraDevine/Temporal_Uncertainty_DevineCA_2019; copy archived at https://github.com/elifesciences-publications/Temporal_Uncertainty_DevineCA_2019). (*Devine, 2019b*). The task consisted of discrete trials, in which they were required to discriminate the direction (left or right) of a target (tilted grating stimulus) based on a change in the relative contrast between two overlaid grating stimuli (see *Figure 1a*). The experiment consisted of 10 blocks, each containing 50 trials and was conducted in a dark, sound-attenuated room with subjects seated in front of a 51 cm cathode ray tube (CRT) monitor (refresh rate: 100 Hz, 1024 × 768 resolution) at a distance of approximately 57 cm. Stimuli used in this task were created using Psychtoolbox and the experiment was presented using Matlab. The stimuli consisted of two overlaid grating patterns (spatial frequency = 1 cycle per degree) presented in a circular aperture (inner radius = 11 outer radius = 6˚) against a dark grey background (luminance: 65.2 cd/m2). Each grating stimulus was tilted by 45˚ relative to the vertical midline (left tilt = −45˚, right tilt = +45˚). The gratings were 'frequency tagged' in order to allow independent measurement of sensory evidence in favour of both possible choices, with the left-tilted grating flickering at 20 Hz and the right-tilted grating flickering at 25 Hz. On each trial the gratings were phase-reversed and these phase-reversals were pseudorandomly counterbalanced across trials.

Subjects commenced each trial by simultaneously clicking the left and right mouse button. A central fixation dot was presented, followed 200 ms later by the presentation of the overlaid grating stimuli at 50% contrast. The stimuli were held constant for the duration of an initial foreperiod, which, as subjects were informed, varied pseudorandomly from trial to trial between 800 ms, 1200 ms and 1600 ms. After the foreperiod, the gratings underwent opposite changes in contrast (the magnitude of which were determined individually using a staircase procedure - see *Staircase Procedure* below) whereby the target stepped up in contrast while the non-target stepped down in contrast by a corresponding amount. The gratings remained at the new contrast level for 2000 ms and subjects were required to indicate the direction of the target stimulus by clicking the corresponding

left or right mouse button with their left or right thumb. Feedback was presented at the end of each trial for 1000 ms and at the end of each block (see *Feedback and Points* below).

## Staircase procedure

The change in contrast of the grating stimuli was determined separately for each subject during the introductory phase of the study, which took place on the first of the six-day study. During this introduction phase, subjects were initially trained to perform the task under relatively easy conditions (large changes in contrast). Once subjects performed the task under easy conditions at close to 100% accuracy, the magnitude of the contrast change was then adjusted downwards gradually over the course of 6–8 short blocks (20 trials each) until subjects were performing within the range of 65–70% accuracy. The average change in contrast resulting from this titration procedure was 5.5% (*SD* = 1.35%, range: 2–7%).

## Feedback and points

At the end of each trial (500 ms post stimulus offset), feedback was presented on screen in text format, for 1000 ms. This feedback indicated whether the subject responded correctly ('Correct'), incorrectly ('Error'), prematurely (responded before or within 150 ms of the contrast change; 'Clicked too Soon'), or failed to respond within the deadline ('Too Late'). At the end of each block, feedback was presented on screen informing subjects of their mean accuracy and response time during the block. This feedback remained on screen until either the subject or experimenter exited the screen. With this feedback, subjects were also given a score for each block reflecting their cumulative points earned during the block. Points were awarded on a trial by trial basis according to the accuracy and speed with which subjects responded. Every correct response was awarded 40 points plus a speed bonus, while incorrect, missed or premature responses were awarded 0 points. The maximum speed bonus was 40 points and this amount diminished linearly from 40 to 0 across the range of possible response times (150–2000 ms). At the end of each block a bar graph was presented to subjects depicting their score for each completed block of the task.

## Experiment 2: random dot motion discrimination task

Subjects performed a difficult, discrete-trial version of the random dot motion discrimination task (*Figure 4a*), which was programmed in Matlab (Mathworks, Natick, MA, USA), using the Psychtoolbox-2 (http://psychtoolbox.org/) package. The task code is available at (https://github.com/CiaraDevine/Temporal_Uncertainty_DevineCA_2019) (*Devine, 2019b*). Subjects were required to judge the direction (left or right) of motion based on a cloud of moving dots. The experiment consisted of 8 blocks, each containing 66 trials and was conducted in a dark, sound-attenuated room with subjects seated in front of a 51 cm CRT monitor (refresh rate: 100 Hz, 1024 × 768 resolution) at a distance of approximately 57 cm. The stimuli used in this task, random dot kinematograms (RDK), were created using Psychtoolbox and the experiment was presented using Matlab. The RDKs consisted of a patch of 100 moving white dots (diameter = 4 pixels) presented within a circular aperture (outer radius = 4°) against a black background. From frame to frame, the dots were displaced throughout the circular aperture creating a perception of motion.

At the onset of each trial, a central fixation dot was presented, followed 400 ms later by the onset of the dot motion stimulus. Initially, the dots were displaced within the aperture at random (0% coherence) giving rise to the perception of random, incoherent motion. The displacement of the dots remained at 0% coherence for the duration of a foreperiod, which as subjects were informed, varied pseudorandomly from trial-to-trial between 800 ms, 1200 ms and 1600 ms. After this foreperiod a portion of the dots began to move coherently in the same direction (either leftward or rightward). This coherent motion was produced by displacing a portion of the dots on each frame, chosen at random, in a common direction relative to their position on the previous frame. All other dots on a given frame were displaced randomly to a new location within the aperture. The percentage of coherently moving dots was determined separately for each subject in the study during a pre-experimental introduction phase (see *'Staircase Procedure'* below). The duration of coherent motion was 2000 ms and subjects were required to indicate its direction by clicking the corresponding left or right mouse button with their left or right thumb. As in experiment one feedback was

presented at the end of each trial for 1000 ms and at the end of each block (see *Feedback and Points* above).

## Staircase procedure

The percentage of coherently moving dots was determined separately for each subject during an introductory phase prior to commencing the experiment. During this phase, subjects were initially trained to perform the task under relatively easy conditions (high coherent motion). Once subjects were proficient at performing the task under easy conditions (were close to 100% accuracy), the difficulty of the task was adjusted using a 2-down 1-up staircase procedure over the course of an 80-trial block that titrated accuracy to 65–70% for each subject individually. Coherence levels ranged from 3–12% ($M$ = 7.42%, $SD$ = 2.61%).

## Motion energy filtering

RDKs are suited to studying the dynamics of sensory evidence accumulation because the random displacement of dots from one trial to the next gives rise to quantifiable variability in sensory evidence within a given trial around the nominal, pre-determined, motion coherence level. In order to estimate the momentary sensory evidence favouring a particular choice we applied dot motion energy filtering (*Adelson and Bergen, 1985*) to the random dot motion stimuli during the pre-evidence foreperiod, using the implementation provided by *Urai and Wimmer (2016)*; see also *Urai et al. (2017)*. This yielded momentary motion energy estimates with positive values reflecting residual rightward motion energy and negative values reflecting residual leftward motion energy. These estimates provided a fine-grained estimate of the momentary sensory evidence favouring the alternative decision outcomes.

## Behavioural analysis

Across both experiments the data were segmented according to foreperiod duration (800/1200/1600 ms). In experiment 1, the data were collapsed across experimental sessions. Response accuracy was defined as the percentage of trials where participants correctly reported the direction of the target. Response time was calculated in milliseconds (ms) relative to the time of the sensory change (i.e. the contrast change or onset of coherent motion). Missed responses were defined as those in which no response was made within the two-second response deadline (marked by the disappearance of the stimulus from the screen). Premature responses were defined as those made prior to or within 150 ms of the onset of sensory evidence (i.e. the contrast change or onset of coherent motion). Points earned per trial were defined as the average number of points earned per trial including trials where no points were awarded due to missed, erroneous or premature responses. Each of these behavioural measures was analysed as a function of foreperiod duration using separate one-way repeated measures analyses of variance (ANOVA) followed by pairwise comparisons (paired-samples t-tests). The alpha level following Bonferroni corrections for each set of pairwise comparisons was. 017 (.05/3). Accuracy was further examined as a function of response time by computing condition accuracy functions. To this end the data were broken down into equally spaced bins based on RT (eight bins in experiment 1 and 6 bins in experiment 2) and analysed using a two-way repeated measures ANOVAs.

In all of the analyses conducted here, prior investigation was carried out to determine whether the data were suitable for parametric analysis. Firstly, the data were screened for any major violations of normality. In the case of repeated measures ANOVAs, Mauchley's test of sphericity was also carried out to assess the assumption of equal variances of the differences across all combined levels of a single factor. In all instances where the assumption of sphericity was violated, greenhouse-geisser corrections were applied to the degrees of freedom and p-values reported.

## The relationship between pre-evidence cumulative motion energy and choice

In light of evidence suggesting that the accumulation process may have commenced before the onset of the goal relevant sensory change, we sought to establish, in experiment 2, whether there was a relationship between random fluctuations in the sensory stimulus during the pre-evidence foreperiod and choice. Specifically, we examined whether fast choices on long foreperiod trials could

be attributed to dot motion energy favouring the chosen response by chance. To this end we expressed momentary motion energy values in terms of whether they favoured the chosen or unchosen response for a given trial. Positive values were attributed to motion energy values favouring the chosen response and negative values were assigned to those favouring the unchosen response. We then divided trials into fast (RT <500 ms) and slow (RT >500 ms) responses and calculated the integral of motion energy during the foreperiod by computing the cumulative sum of the momentary motion energy values expressed with respect to the chosen responses. The cut-off of 500 ms between fast and slow responses was chosen in order to capture a sufficiently large subset of trials in which accuracy was very low, as evident in *Figure 4d*. In the case of responses made after the coherent motion onset, the integral was calculated between 100–1600 ms. For responses made prior to the onset of coherent motion, the integral was calculated between 100 ms and the time of the response. The integral at 1600ms and the slope of the pre-evidence integral (measured as the slope of a line fit to the pre-evidence motion integral) for fast and slow responses was compared against 0 using single sample t-tests. Whereas the absolute value of the integral at 1600 provides a measure of how much evidence cumulatively favoured the chosen response, the slope of the integral is sensitive to the extent to which sensory evidence consistently favoured the chosen response.

## EEG acquisition and preprocessing (experiments 1 and 2)

The following procedures for acquiring and preprocessing EEG data apply to data from experiment 1 and experiment 2. Continuous EEG data were acquired using an ActiveTwo system (BioSemi, The Netherlands) from 128 scalp electrodes and digitized at 512 Hz. Vertical eye movements were recorded using two vertical electrooculogram (EOG) electrodes placed above and below the left eye. Data were analysed using custom scripts (available at https://github.com/CiaraDevine/Temporal_Uncertainty_DevineCA_2019) in MATLAB (Mathworks, Natick, MA) and the EEGLAB toolbox (*Devine, 2019b*; *Delorme and Makeig, 2004*). The full datasets for experiment 1 and experiment two can be found at https://doi.org/10.5061/dryad.b2rbnzs8r (*Devine, 2019a*). Continuous EEG data were low-pass filtered below 35 Hz, high-pass filtered above. 05 Hz and detrended. EEG data were then re-referenced offline to the average reference. EEG data were segmented into stimulus- and response-aligned epochs. Stimulus-aligned epochs were extracted from stimulus onset to stimulus offset, which yielded different length windows for each foreperiod condition: 800 ms (Epoch: 0–2,800 ms), 1200 ms (Epoch: 0–3,200 ms) or 1600 ms (Epoch: 0–3,600 ms). Response-aligned epochs were measured from –1000 ms pre-response to 600 ms post response. The purpose of extending the epoch 600 ms post-response was to allow four full fast fourier transform (FFT) windows to be calculated post-response (*see Time Frequency Decomposition below*). All epochs were baseline-corrected relative to the interval of 500–550 ms post stimulus onset. This baseline window was chosen, with the aid of visual inspection of the grand-average event-related potential (ERP) waveform, to fall before the onset of the CPP but after the conclusion of evoked potentials elicited by the appearance of the stimulus on the screen and the mouse click performed by subjects in order to initiate each trial.

Trials were rejected if the bipolar vertical EOG signal (upper minus lower) exceeded an absolute value of 200 µV or if any scalp channel exceeded 100 µV at any time during the stimulus-aligned epoch. To avoid excessive trial loss, channels were interpolated if their individual artefact count exceeded 10% of the total number of trials for a given session. To avoid excessive channel interpolation, a maximum of 10% of the total number of channels were permitted to be interpolated for any given subject's data in a given session. Subjects were excluded from electrophysiological analyses entirely if, following these steps, more than 40% of trials were lost due to blinks and/or EEG artefacts. In order to mitigate the effects of volume conduction across the scalp, single-trial epochs underwent current source density (CSD) transformation (*Kayser and Tenke, 2006*), a procedure that helps to minimise spatial overlap between functionally distinct EEG components (*Kelly and O'Connell, 2013*), an issue that may confound our interpretation of key signals such as the CPP (*Philiastides et al., 2014*).

Stimulus- and response-aligned epochs were then decomposed into time frequency representations using the Short Time Fourier Transform (STFT) procedure. Each epoch was divided up into a series of overlapping 400 ms time segments taken at 50 ms intervals across the epoch. The FFT was computed for each 400 ms segment. For the purpose of data analysis and plotting, each 400 ms

segment was identified by its median time point. The resulting time-frequency representations had a frequency resolution of 2.5 Hz and as such allowed for the direct measurement of SSVEPs at 20 Hz and 25 Hz, which corresponded to the flicker frequencies used in experiment one for the left- and right-tilted gratings respectively. Each 400 ms window captured precisely 8 and 10 cycles of the 20 and 25 Hz SSVEPs respectively, thereby minimising spectral leakage. FFT power measurements were converted into amplitude measurements by dividing by half the length of the measurement window in samples.

## Analysis of electrophysiological signals

### The effect of foreperiod duration on sensory evidence representation

Sensory evidence representation was examined by tracing the time course of the SSVEPs elicited by the left- and right-tilted stimuli flickering at 20 Hz and 25 Hz respectively. The SSVEPs were normalised with respect to the average activity in the immediately adjacent frequencies by computing signal to noise ratios (SNR). In order to obtain a direct measurement of the sensory evidence upon which observers based their decisions (i.e. the relative contrast of the overlaid gratings) we then computed the difference SSVEP (d-SSVEP) by subtracting the non-target SSVEP from the target SSVEP. Based on visual inspection of the sample grand-average topography of the d-SSVEP (calculated at approximately 200 ms prior to response) we identified a broad cluster of electrodes over the occipital cortex (centred around Oz) at which the d-SSVEP was maximal (*Figure 2a*). At the single-subject level we then measured the SSVEPs by averaging data from the four electrodes within this cluster, at which the subject grand-average d-SSVEP was maximal. Single trial data were rejected from subsequent analyses if the d-SSVEP prior to response exceeded + /- 3 standard deviations from the within subject mean.

To determine whether foreperiod duration had any effect on sensory evidence representation the SSVEPs were examined as a function of foreperiod duration in the following windows: i) 500 ms post evidence onset (the earliest time-point at which the SSVEP SNR has already reached a plateau) and ii) 200 ms prior to response (The time-point closest to response at which there would be little or no effect of post-response attentional disengagement). Trials were excluded from this analysis if subjects responded quicker than 500 ms in order to ensure that response-aligned SSVEP measurements did not include time windows prior to the contrast change and that post evidence measurements overlapped minimally with time points post response. This analysis was carried out using two-way repeated measures ANOVAs (factor 1: foreperiod duration; factor 2: stimulus). Effects of foreperiod duration on the d-SSVEP were inferred from the interaction between foreperiod duration and stimulus factors. We predicted that the target and non-target SSVEPs should undergo antithetical changes in SNR after the contrast change giving rise to a main effect of stimulus.

### The effect of foreperiod duration on pre-evidence decision formation

Domain-general decision formation was examined by measuring the dynamics of the CPP. Based on visual inspection of the sample grand-average topography of the pre-response CPP (response aligned window: −150 ms to −50 ms) we identified a broad cluster of electrodes over the centroparietal cortex (centred around Pz) at which the CPP was maximal (*Figure 3a*; 6a). For each subject, the CPP was measured by averaging data from the four electrodes within this cluster, at which the subject grand-average was maximal. Effector selective decision formation was examined by measuring desynchronisation in the Mu/Beta frequency band (10–30 Hz) over the contralateral and ipsilateral response hemispheres (20 Hz and 25 Hz were excluded in order to avoid mixing motor and sensory activity). In order to measure its excursion and to minimise the effects of general changes in mu/beta amplitudes across testing sessions or between trials, Mu/Beta was baseline-corrected relative to 550 ms post stimulus onset. This window was chosen with the aid of visual inspection of the grand average waveforms to ensure that the motor activity relating to the mouse button click at the beginning of each trial in experiment one was fully resolved. The Mu/Beta lateralisation index was computed by subtracting ipsilateral from contralateral Mu/Beta. Based on inspecting the grand average topography of pre response Mu/Beta lateralisation (response aligned window: −150 to −50 ms), a cluster of electrodes was identified in each hemisphere over premotor regions of the scalp (C3 left hemisphere; C4 right hemisphere; *Figure 3h*; 6 hr) that exhibited the largest lateralisation index. For each subject Mu/Beta was measured by averaging data from the four electrodes within this cluster,

at which the subject grand-average was maximal. Single trial CPP and Mu/Beta data were rejected from subsequent analyses if the values estimated prior to response exceeded + /- 3 standard deviations from the within-subject mean.

Inspection of grand average CPP and lateralised Mu/Beta waveforms indicated that there was substantial build-up of both signals during the foreperiod on longer foreperiod trials suggesting that decision formation may have commenced prematurely on those trials. To investigate this phenomenon we initially pooled data from correct, incorrect, missed and premature response trials and separated the data according to foreperiod duration. We examined the amount of pre-evidence build-up in each signal by calculating the average amplitude of each signal within a 100 ms window centred on the time of the sensory change. This window was selected so as to capture the full extent of the pre-evidence build-up while avoiding overlap with activity that could feasibly be driven by the sensory change. We also measured the rate of build-up during the foreperiod as the slope of a line fit to the waveforms between −250 ms and 50 ms relative to the sensory change. This window was selected to ensure that it did not reach back before the onset of the CPP (which would result in underestimation of the slope) and, in the case of short FP trials, to ensure that there was no overlap with visual-evoked potentials driven by the stimulus onset (500 ms prior to evidence onset). Nevertheless, after repeating the analyses using wider time windows for the pre-evidence slope measurement −550 to 50 ms and −950 to 50 ms windows), we found that the key trends presented in *Figure 3f* were unchanged. In order to determine whether there was significant pre-evidence decision related activity, the slope and amplitude of CPP and Mu/Beta lateralisation at each level of foreperiod duration were compared against 0 using one-sample t-tests. We repeated this analysis for long foreperiod trials using only data from experimental session 1. Next, we divided the data into equally sized bins based on RT (six bins in experiment 1; three bins in experiment 2) and examined the relationship between RT and pre-evidence decision related activity (amplitude and slope as defined above) using two-way repeated measures ANOVAs including foreperiod duration and RT bin as factors.

## The effect of foreperiod duration on pre-response decision related activity

Decision formation dynamics were further investigated relative to response. We measured the amplitude of the CPP and magnitude of Mu/Beta desynchronisation and lateralisation prior to response in the window −150 ms to −50 ms. This window was selected in light of the assumption that there is a time lag between decision commitment and response execution. Here we centred the measurement window on −100 ms as this coincides with the onset timing of the sharp pre-response deflection in the motor readiness potential (*Steinemann et al., 2018*). Contralateral and ipsilateral Mu/Beta were examined separately so as to evaluate preparation of both the chosen and unchosen responses. The build-up rate of the CPP was measured by calculating the slope of a line fit to the waveform in the window −500 ms to −200 ms relative to response. The build-up rate of Mu/Beta lateralisation was examined by measuring the slope of a line fit to the lateralisation indices contained within the window −500 ms to −200 ms relative to response. The pre-response slope measurement windows were chosen so as to ensure that the estimates of build-up rate were determined during the pre-commitment time period and to avoid the possibility that the slope would be underestimated as a result of an excessively long time window that extended to pre accumulation onset. In particular there was a very strong likelihood that by using a wider pre-response slope measurement window on longer foreperiod trials that the slope estimate would be based on pre-evidence accumulation as opposed to post evidence accumulation. Again, we did not find any change in the key trends (*Figure 3g*) reported on short  er foreperiod trials as a result of using a wider pre-response slope measurement window (e.g. −900 ms to −200 ms and −1200 ms to −200 ms). As before, the data were separated according to foreperiod duration and RT and analysed using two-way repeated measures ANOVAs.

## Examining the relationship between motion energy and premature sensory evidence accumulation (experiment 2)

In experiment two we sought to establish a relationship between cumulative motion energy during the foreperiod and pre-evidence CPP build-up. To this end, focusing exclusively on long foreperiod (1600 ms) trials, we separated trials into two equally sized bins according to the slope of the dot motion integral during the foreperiod. This yielded a subset of trials in which the cumulative motion

energy during the foreperiod strongly and consistently favoured either leftward or rightward motion by change and a subset of trials where the cumulative motion energy fluctuated closer to 0, favouring neither direction consistently. Single sample t-tests were first carried out, comparing the pre-evidence CPP amplitude and slope against 0, in order to determine whether there was any significant build-up of the CPP during the foreperiod on steep or shallow integral trials. The amplitude of the CPP was again measured in a 100 ms window centred on coherent motion onset while the slope was measured in a broad window of 600 ms to 1600 ms. This window was again selected so as to avoid any overlap with early visual evoked potentials. By focusing this analysis on 1600 ms foreperiod trials we were able to estimate the pre-evidence CPP slope over a longer period of time. To distinguish between the CPP on steep and shallow pre-evidence integral trials, additional paired samples t-tests were carried out comparing the amplitude and slope of the CPP between shallow and steep integral trials.

## Examining the effect of previous trial foreperiod on accumulation onset timing

In order to examine the sequential effect of foreperiod on evidence accumulation, the single trial data from experiment were subsequently further subdivided according to the duration of the foreperiod on the previous trial (foreperiod$_{n-1}$) and the duration of the foreperiod on the current trial (foreperiod$_n$). At the behavioural level accuracy and response times were examined using two-way repeated measures ANOVAs including foreperiod$_{n-1}$ and foreperiod$_n$ as separate independent factors. Missed response rate and premature response rate were examined using one-way repeated measures ANOVAs including only foreperiod$_{n-1}$ as an independent factor. The exclusion of foreperiod$_n$ as a factor in these analyses was necessary because missed responses were made almost exclusively on short foreperiod$_n$ trials while premature responses were made almost exclusively on long foreperiod$_{n-1}$ trials. At the neural level, we also further analysed the CPP as a function of foreperiod$_{n-1}$ in order to determine whether the amount or rate of build-up, in either the pre-evidence onset or pre-response windows, was modulated by the duration of the previous trial foreperiod.

## Examining target selection signals under temporal uncertainty

The target selection process was examined by measuring the bilateral N2 which is thought to play a role in triggering the onset of evidence accumulation (*Loughnane et al., 2016*). Based on visual inspection of the sample grand-average topography of the ERP in the window 200 ms to 300 ms post sensory change we identified a broad cluster of electrodes over the bilateral occipital cortex at which negative going activity was maximal. For each subject, the N2 was measured by averaging data from three electrodes centred on P7 and 3 electrodes centred on P8 at which the negativity was greatest. To establish whether target selection mechanisms were reliably elicited by the sensory change in this task we compared the amplitude of the N2, measured in the window 200 ms to 300 ms post sensory change, against 0 using separate single sample t-tests for each level of foreperiod.

## Additional information

### Funding

| Funder | Grant reference number | Author |
| --- | --- | --- |
| Irish Research Council | Postgraduate Fellowship | Ciara A Devine<br>Redmond G O'Connell |
| H2020 European Research Council | Starting Grant 63829 | Redmond G O'Connell |
| National Science Foundation | BCS-1358955 | Simon P Kelly<br>Redmond G O'Connell |

The funders had no role in study design, data collection and interpretation, or the decision to submit the work for publication.

## Author contributions
Ciara A Devine, Conceptualization, Software, Formal analysis, Funding acquisition, Investigation, Visualization, Methodology, Project administration; Christine Gaffney, Investigation, Project administration; Gerard M Loughnane, Formal analysis; Simon P Kelly, Conceptualization, Software, Methodology; Redmond G O'Connell, Conceptualization, Resources, Software, Supervision, Funding acquisition, Methodology

## Author ORCIDs
Ciara A Devine https://orcid.org/0000-0001-7522-1172
Gerard M Loughnane http://orcid.org/0000-0003-1961-5294
Simon P Kelly http://orcid.org/0000-0001-9983-3595
Redmond G O'Connell https://orcid.org/0000-0001-6949-2793

## Ethics
Human subjects: Written, informed consent was obtained from all subjects prior to taking part in this study and all procedures were approved by the Trinity College Dublin ethics committee (SPREC112014-01) and conducted in accordance with the Declaration of Helsinki.

## Decision letter and Author response
Decision letter https://doi.org/10.7554/eLife.48526.sa1
Author response https://doi.org/10.7554/eLife.48526.sa2

# Additional files
## Supplementary files
• Transparent reporting form

## Data availability
Data is available on dryad at https://doi.org/10.5061/dryad.b2rbnzs8r and Github https://github.com/CiaraDevine/Temporal_Uncertainty_DevineCA_2019 (copy archived at https://github.com/elifesciences-publications/Temporal_Uncertainty_DevineCA_2019).

The following dataset was generated:

| Author(s) | Year | Dataset title | Dataset URL | Database and Identifier |
|---|---|---|---|---|
| Devine CA | 2019 | The Role of Premature Evidence Accumulation in Making Difficult Perceptual Decisions under Temporal Uncertainty | https://doi.org/10.5061/dryad.b2rbnzs8r | Dryad Digital Repository, 10.5061/dryad.b2rbnzs8r |

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
