## [Decision Letter]

**Acceptance summary:**

Your revisions have helped provide a richer understanding of the originally reported effects on the role of pre-evidence build-up of information as captured by the CPP. As such your work makes an important contribution to our understanding of the neural correlates of information accumulation in the presence of temporal uncertainty in the onset of the relevant decision evidence.

**Decision letter after peer review:**

Thank you for submitting your article "The role of premature evidence accumulation in making difficult perceptual decisions under temporal uncertainty" for consideration by *eLife*. Your article has been reviewed by three peer reviewers, including Marios Philiastides as the Guest Editor and Reviewer #3, and the evaluation has been overseen by Michael Frank as the Senior Editor. The following individual involved in review of your submission has agreed to reveal their identity: Laurence Tudor Hunt (Reviewer #1.

The reviewers have discussed the reviews with one another and the Guest Editor has drafted this decision to help you prepare a revised submission.

Summary:

Most laboratory studies of (perceptual) decision making are designed such that there is little ambiguity as to when participants need to start integrating information for the decision. The main aim of this study was to study how information accumulation is initiated when the temporal onset of the relevant information is temporally uncertain. The authors used human psychophysics and electrophysiology to shed some light on this issue. The question is interesting and timely and the paper is generally well-motivated and well-written.

Essential revisions:

1) Three accounts were presented in the Introduction as possible mechanisms for how the process of evidence accumulation is initiated (gated accumulator model, early target selection, temporal trade-off based on temporal structure of the task). The paper argues that the latter account might be more likely given the behavioural and neural observations but it stops short from formally arbitrating between the possible mechanisms, for example, by building and comparing generative models of the various alternatives. This exercise would help validate the main conclusions of the paper. At the very least the new analyses suggested below should be used to further tease these accounts apart.

2) Some of the most relevant effects (e.g. amplitude/slope of CPP as a function of RT) failed to replicate in Experiment 2, which in turn challenges the general nature of the reported effects (i.e. could this, instead, be due to differences in the type of sensory evidence across the two experiments). The authors state this is due to low power. Given the reproducibility crisis in the field, this point needs to be rectified, possibly with the addition of more data. (Please let us know if you do decide to collect more data and would like to request additional time for revision).

3) One concern is the missing link between the origin of the actual sensory signals and the proxy signatures of evidence accumulation as detected by the CPP. For instance, do the trial-to-trial fluctuations in the SSVEPs before trial onset already play a role in biasing the decision (CPP) signals? Similarly, it remains unclear whether premature signals have a sensory origin alone (i.e. primarily due to motion energy), or perhaps are related to motor biases due to experiences of previous trials (including correct, incorrect choices, temporal structure, etc). For instance, it is possible that pre-trial absolute levels of mu/beta activity in the left and right hemispheres (or lateralization index) cause early drifts in evidence accumulation. Similarly, temporal structure can also have an impact on the slope of evidence accumulation. It would be beneficial to investigate which of these variables explain the most variance in the reported neural and behavioural effects.

4) More broadly, what is the relationship between the CPP and motor preparation signals? Is one derived from the other? Is there some sort of meaningful (functionally significant) trial-by-trial association? Currently there is a disconnect between these two signals and their relative contributions to the observed choices.

5) By design, the experiment has a number of trials where the subjects respond incorrectly and/or miss the trial altogether. Would it be possible to look at these trials separately, and examine how the CPP and lateralised motor responses vary as a function of making a correct response versus making an error? Do the EEG responses during the foreperiod have different properties on error trials than correct trials? The current approach (combining all trials) is potentially masking interesting interactions between accuracy and foreperiod duration (as Figure 1B/4B suggest, correct and error RTs are quite similar in the 1600ms but different in the 800ms condition).

6) There is a potential limitation of only examining trials which have a relatively limited foreperiod (only up to 1.6s). This means that subjects can reasonably infer that they will be in a trial within a certain period after foreperiod onset. How would this change in situations where the foreperiod is far longer than this, and so subjects really cannot estimate when a trial begins or ends? Which of the 'foreperiod' EEG signals could still be looked for in a design with far longer foreperiods? Following from this, the CPP build-up appears to develop roughly at the same time point (~800ms) across all three conditions, consistent with a strategy whereby participants calibrate their onset of evidence accumulation to the shortest foreperiod (i.e. inconsistent with the regression to the mean interpretation discussed in the paper). This should be discussed explicitly along with the caveat that these effects might be specific to the intricacies of the task used here.

7) Methodological Considerations: The selection of the windows over which the CPP/Mu/Beta slopes and amplitudes were estimated in the two experiments is somewhat arbitrary. How were these decided and how robust are the results relative to the choice of these windows? Given the slow nature of the CPP signal (unfolding over a second or more) it would seem that a longer window might be more appropriate to accurately estimate the slope. Relatedly, the choice of a linear line fit needs to be better motivated since the CPP appears to build-up quadratically. Baseline correction might be masking important neural signatures of decision formation in the absence of sensory evidence (i.e. pre-trial influences). Do the main effects presented in the paper persist when CPP, mu/beta and lateralization estimates are derived without any baseline correction?

8) Treating the literature: There is a somewhat biased sampling of the literature. There are multiple groups that have provided evidence on the role of CPP in decision making using human electrophysiology and ideally would need to be acknowledged appropriately.

9) Data availability: Data sharing is central to e*Life*'s mission as well as the ERC ("FAIR data principles") funding this work. Fully anonymised data from normal control groups as in this study (raw data or subject-wise IVs required to reproduce the findings presented in the main figures) should be shared openly.

[Editors' note: further revisions were requested prior to acceptance, as described below.]

Thank you for resubmitting your work entitled "The role of premature evidence accumulation in making difficult perceptual decisions under temporal uncertainty" for further consideration by *eLife*. Your revised article has been evaluated by Michael Frank as the Senior Editor, Marios Philiastides as the Guest Editor, and one reviewer.

The manuscript has been improved but there are some remaining issues that need to be addressed before acceptance, as outlined below:

[Original comment #3] As reviewer 2 highlights below, the question of how baseline visual or motor activity contributes to biases in evidence accumulation (which can offer mechanistic insights into the role of premature accumulation) has not been fully addressed.

We understand that unlike Experiment 2, in Experiment 1, participants made simultaneous left/right button presses to initiate a trial which are likely to contaminate the pre-stimulus period. One possibility to address this issue would be to define a new "baseline" during the presentation of the 50% contrast (ambiguous) stimulus. Since the earliest onset time of the accumulation signal in the current data appears around 800ms after the ambiguous stimulus is presented, this could serve as an objective cutoff for defining a new pre-accumulation baseline window.

[Original comment #6] Please discuss the implications of short-vs.-long foreperiods explicitly, along the lines of what was highlighted in your response to the reviewers. That is, the effect of premature accumulation is likely to manifest only under certain conditions (e.g. when rapid decisions are required).

[Original comment #9] Data sharing: *eLife*'s list of recommended repositories can be found here: https://fairsharing.org/bsg-p000124/.

There are a number of repositories on the list (e.g. Openneuro, Datadryad etc) that can accommodate the type and size of your data. Please add a link to your shared data in your revised manuscript.

Reviewer #2:

This is a revised version of the manuscript, where the authors have addressed most the reviewer comments. My main criticism (related to the summarized reviewer comment #3 in the first round of reviews) was that there is little insight on what causes biases in the decision (CPP) signals, issue that was not addressed by the authors. I think this could have been partially addressed based on the acquired EEG data. For instance, it could be examined whether baseline visual or motor activity contribute to evidence accumulation biases, and how much this affects physical input information (as done in Experiment 2) beyond what is known of CPPs from previous studies by the authors. In my opinion, this information could help to understand the current data and findings a bit more "mechanistically" beyond the more descriptive domain.

We will look forward to hearing from you with a revised article and a response letter describing the changes made.

---

## [Author Response]

Essential revisions:1) Three accounts were presented in the Introduction as possible mechanisms for how the process of evidence accumulation is initiated (gated accumulator model, early target selection, temporal trade-off based on temporal structure of the task). The paper argues that the latter account might be more likely given the behavioural and neural observations but it stops short from formally arbitrating between the possible mechanisms, for example, by building and comparing generative models of the various alternatives. This exercise would help validate the main conclusions of the paper. At the very least the new analyses suggested below should be used to further tease these accounts apart.

As we describe below, the additional analyses suggested by the reviewers do indeed bolster our contention that evidence accumulation is initiated not by the evidence representation itself or the detection of its onset, but rather in a self-timed manner in advance of the evidence onset. Our data thus give strong qualitative grounds for adjudicating between these three accounts – since the first two entail no evidence accumulation until some post-evidence event triggers it, the empirical evidence for pre-evidence accumulation rules them out for this task. Ultimately, it will indeed be important to construct models in future work that can account for the full range of behavioural and neural observations in the present data in addition to the foreperiod effects, which will require careful consideration of a number of potentially interacting factors including the role of static and dynamic urgency adjustments and drift rate variability.

2) Some of the most relevant effects (e.g. amplitude/slope of CPP as a function of RT) failed to replicate in Experiment 2, which in turn challenges the general nature of the reported effects (i.e. could this, instead, be due to differences in the type of sensory evidence across the two experiments). The authors state this is due to low power. Given the reproducibility crisis in the field, this point needs to be rectified, possibly with the addition of more data. (Please let us know if you do decide to collect more data and would like to request additional time for revision).

We thank the reviewers for this comment which has prompted us to reconsider and re-examine the relationship between pre-evidence CPP accumulation and RT across the two experiments.

In fact, an overall inverse relationship between pre-evidence accumulation and RT should not necessarily be expected: whereas pre-evidence accumulation should lead to faster responses when it happened to favour the ultimately chosen alternative, it would also cause slower responses when it happened to favour the ultimately unchosen alternative because it would serve to push the decision process further from its final bound. Thus, the degree to which a statistically significant inverse relationship between pre-evidence CPP and RT would manifest would depend on the prevalence of this latter subset of ‘change of mind’ trials. This led us to further interrogate the CPP-RT relationship observed in Experiment 1 with some new analyses. First, because such changes of mind from the pre- to post-evidence period should be far more prevalent on correct trials, we examined the relationship between RT and pre-evidence CPP amplitude separately for correct and error trials. As expected, we found that the relationship was significantly stronger on error trials than on correct ones (repeated measures ANOVA RT bin x Choice (correct vs. error) interaction: F(3, 54)=3.94, p=.02; Figure 3—figure supplement 2B). Second, we discovered that, because of the nature of the stimulus used in Experiment 1, pre-evidence accumulation was systematically biased in favour of left choices. Subjects were 4.17 times more likely to choose left when responding prematurely (<150ms; t(18)=3.20, p=.005; Figure 1—figure supplement 1A), 1.95 times more likely to choose left when responding quickly (<500ms; t(20)=5.62, p=.1.69^-4^; Figure 1—figure supplement 1B) and had overall faster RTs when left targets were presented (F(1, 20)=63.65, p=1.22^-7^; Figure 1—figure supplement 1C). This systematic bias can be attributed to the difference in flicker frequency between left (20Hz) and right-tilted (25Hz) gratings. Psychophysical research has previously shown that within this range the perceived brightness (Bartley, 1938; 1951; Wu, Burns, Reeves and Elsner, 1996) and contrast (Solomon and Tyler, 2018) of flickering stimuli diminishes as a function of flicker frequency, even when they are matched in terms of physical contrast and luminance. Since the left-tilted grating appeared higher-contrast during the foreperiod despite physical equivalence, this may have caused pre-evidence accumulation to be biased toward left choices. Consequently, there are likely to be far more change of mind trials when “right” was the correct response than when “left” was the correct response. Correspondingly, we found that the relationship between pre-evidence CPP and RT was stronger for left than right choice trials (RT x choice interaction: F(3, 54)=3.19, p=.03; Figure 3—figure supplement 2A).

In contrast, in Experiment 2 there was no significant choice bias on premature response trials (Figure 4—figure supplement 1A-B; Ratio of Left to Right choices =.88; t(16)=-.69, p=.50; subjects with fewer than 10 premature choices were excluded) and no overall RT bias (Figure 4—figure supplement 1C; Left Responses: *M*=998ms, *SD*=89.28ms;Right Responses: *M*=1091ms, *SD*=117.37ms; F(1, 21)=.94, p=.34). This indicates that pre-evidence accumulation was not systematically biased in experiment two. Thus we conclude that the most likely explanation for the discrepant CPP-RT results across the two experiments is the presence of a choice bias in Experiment 1 which caused a reduction in the number of slow change of mind trials and ensured that pre-evidence accumulation was more likely to accelerate, rather than delay, responses. There is nothing in our data to suggest that this choice bias impacted substantially on the participants’ decision making strategies or on the timing or extent of pre-evidence accumulation – participants were made fully aware of the foreperiods during the initial training for both experiments (including an initial set of trials for which the evidence onsets were highly salient to render the zero evidence foreperiods more obvious) and the distinct foreperiods had highly similar effects on RT distributions, choice accuracies and CPP build-ups across the two experiments. We are grateful to the reviewers for bringing this matter to our attention. These discussion points and empirical observations are now included in the revised manuscript (see subsections “Premature Decision Formation on Trials with Longer Foreperiods” and “Decision Formation Commenced Prematurely on Longer Foreperiod Trials and Was Predictive of Behaviour”) alongside relevant figure supplements.

3) One concern is the missing link between the origin of the actual sensory signals and the proxy signatures of evidence accumulation as detected by the CPP. For instance, do the trial-to-trial fluctuations in the SSVEPs before trial onset already play a role in biasing the decision (CPP) signals? Similarly, it remains unclear whether premature signals have a sensory origin alone (i.e. primarily due to motion energy), or perhaps are related to motor biases due to experiences of previous trials (including correct, incorrect choices, temporal structure, etc). For instance, it is possible that pre-trial absolute levels of mu/beta activity in the left and right hemispheres (or lateralization index) cause early drifts in evidence accumulation. Similarly, temporal structure can also have an impact on the slope of evidence accumulation. It would be beneficial to investigate which of these variables explain the most variance in the reported neural and behavioural effects.

The reviewers highlight an important question in asking to what extent the build-up of the decision signals, such as the CPP, directly reflect the integration of sensory evidence alone as opposed to other factors. We would argue that the key contribution of our paper is to show that the CPP’s build-up is at least in part driven by sensory evidence in Experiment 2 which is most compellingly manifested in its sensitivity to cumulative residual dot motion energy in the pre-evidence period. This observation provides some of the first empirical data supporting the claim that accumulation onset timing is a strategically modifiable parameter of the decision process (Teichert et al., 2016). We have previously demonstrated that fluctuations in the SSVEP do indeed influence the CPP (O’Connell et al., 2012) and would expect similar relationships to exist here but the motion energy analyses provide a more powerful test because they do not rely on noisy single-trial SSVEP measurements, the temporal resolution of which is poor (each estimate is based on a 400ms FFT window). While we believe our results provide a convincing demonstration that the early build-up of the CPP at least partly reflects an early accumulation of sensory information, we certainly do not exclude the possibility that the CPP is subject to other influences. For example, with respect to temporal structure, we have indeed demonstrated in Experiment 1 that the onset time of the CPP is highly sensitive to the foreperiod of the previous trial. This is an important and intriguing result because it highlights that decision onset times were not fixed (e.g. to coincide with stimulus onset) but continually adjusted in light of recent events. It is very likely that additional factors such as choice history biases and attentional fluctuations also affect the CPP’s build-up but while quantifying such contributions has been a focus of our own research and that of other groups, this particular matter does not appear to bear on the central claim of the current paper. We thank the reviewers for highlighting a lack of clarity in our articulation of the key contributions of this paper. We have now made amendments to the text in order to address this (see Discussion).

4) More broadly, what is the relationship between the CPP and motor preparation signals? Is one derived from the other? Is there some sort of meaningful (functionally significant) trial-by-trial association? Currently there is a disconnect between these two signals and their relative contributions to the observed choices.

We thank the reviewer for highlighting that we had not provided background of sufficient depth to explicate how the two signals most likely relate to each other, and the recent evidence that bears on this matter. We have thus now given a more complete account of the known functional characteristics of the CPP and mu/beta decision signals in the Introduction. The reviewer correctly points out that the precise relationship between the CPP and motor preparation signals is yet to be fully determined, but previous work has shed some light on this matter. First, the evidence-dependent build-up of the CPP has been shown to reliably precede that of motor preparation signals (Kelly and O’Connell, 2013). Second, it has been shown across several studies that the CPP and motor signals undergo qualitatively distinct strategic adjustments: premotor Mu/Beta-band activity contralateral to the decision reporting effector always reaches a stereotyped threshold level prior to response execution but both contralateral and ipsilateral signals exhibit systematic shifts in their starting levels in response to prior information about time constraints (Steinemann et al., 2018) and stimulus probability (Kelly et al., 2019), as well as a temporally increasing urgency component to their build-up (Kelly et al., 2019; Murphy et al., 2016; Steinemann et al., 2018). In contrast, in two recent studies it was observed that the CPP did not change its starting level and its pre-choice amplitude varied systematically as a function of RT for discrete decisions with a time limit (Kelly et al., 2019; Philiastides et al., 2014; Steinemann et al., 2018). Together, these data suggest that the CPP encodes a pure representation of cumulative evidence which is fed to the motor level and combined with other strategic influences. Though more precisely establishing the functional relationship between mu/beta and the CPP is an interesting and important avenue for further research we would argue that the CPP has been sufficiently characterised to support our interpretation of the key results of this paper.

5) By design, the experiment has a number of trials where the subjects respond incorrectly and/or miss the trial altogether. Would it be possible to look at these trials separately, and examine how the CPP and lateralised motor responses vary as a function of making a correct response versus making an error? Do the EEG responses during the foreperiod have different properties on error trials than correct trials? The current approach (combining all trials) is potentially masking interesting interactions between accuracy and foreperiod duration (as Figure 1B/4B suggest, correct and error RTs are quite similar in the 1600ms but different in the 800ms condition).

As the reviewer suggests, separating trials according to response accuracy does in fact reveal some interesting interactions, most notably (addressed in response to reviewer comment 2), the interaction between RT and choice accuracy with regard to pre-evidence accumulation on long foreperiod trials (RT x choice interaction: F(3, 54)=3.94, p=.02; Figure 3—figure supplement 2B). Specifically we see that the linear relationship between RT and pre-evidence accumulation is much stronger on error trials, an observation which can be attributed to the fact that ‘changes of mind’ (where pre-evidence accumulation initially favours one alternative but the physical evidence then favours the other) responses are far more likely on correct response trials and will counteract the otherwise inverse relationship between pre-evidence CPP and RT.

6) There is a potential limitation of only examining trials which have a relatively limited foreperiod (only up to 1.6s). This means that subjects can reasonably infer that they will be in a trial within a certain period after foreperiod onset. How would this change in situations where the foreperiod is far longer than this, and so subjects really cannot estimate when a trial begins or ends? Which of the 'foreperiod' EEG signals could still be looked for in a design with far longer foreperiods? Following from this, the CPP build-up appears to develop roughly at the same time point (~800ms) across all three conditions, consistent with a strategy whereby participants calibrate their onset of evidence accumulation to the shortest foreperiod (i.e. inconsistent with the regression to the mean interpretation discussed in the paper). This should be discussed explicitly along with the caveat that these effects might be specific to the intricacies of the task used here.

We agree that it is likely that the phenomenon of premature accumulation will only manifest under certain conditions. In a previous study from our lab (Loughnane et al., 2016) subjects performed a continuous version of the dot motion discrimination task in which the inter-target interval varied between 3-7.3 seconds but coherence levels were set at much higher levels (35-45%). At these higher evidence strengths we found that, the CPP showed no sign of premature onset but was reliably preceded, within 200ms following coherent motion onset, by a sensory-driven target selection signal (N2). In the current study, however, our data indicate that, due to the weaker evidence, subjects were unable to rely on such target selection processes in order to accurately align accumulation onset to the timing of the sensory change, and had to instead time their accumulation onset with respect to stimulus onset despite the variable timing in the sensory change thereafter. It would indeed be very interesting to expand the range of foreperiod durations and to examine evidence accumulation processes. Such a study, and the examination of all of the decision signals considered in the current paper would be entirely feasible. For now, we feel that the present paper makes an important contribution by drawing attention to the fact that evidence accumulation onset time is an endogenously set parameter of the decision process and that the CPP provides access to these adjustments.

Secondly, we welcome the reviewer highlighting that regression to the mean is not the only plausible interpretation of the strategy deployed in our task with respect to timing the onset of accumulation and we are glad to have the opportunity to revise and clarify this point (see Discussion). As per the reviewers comments, it is clear based on the CPP that subjects timed the onset of evidence accumulation to approximately coincide with the earliest possible time point at which evidence could be presented to them (800ms). Interestingly, rather than optimising overall performance on shorter foreperiod trials (which had a notably high rate of missed responses and earned the lowest number of points), this strategy instead led to a peak in performance on intermediate foreperiod trials (which had notably high levels of accuracy and low levels of both missed and premature responses thus earning a higher number of points). One speculative explanation for this is that, despite timing accumulation onset relative to the shortest foreperiod, in estimating the response deadline, subjects may have relied on the bayesian approach, ‘regression to the mean’, which is deployed in estimating uncertain time intervals (Jazayeri and Shadlen, 2010; 2015). Given that the foreperiod durations used in this experiment were drawn from a uniform distribution, this strategy would cause subjects to bias their timing estimates of the impending deadline towards the trial duration on intermediate foreperiod trials. This is an interesting avenue for further investigation and could be examined by repeating this study, drawing the foreperiod duration from various asymmetrical distributions.

7) Methodological Considerations: The selection of the windows over which the CPP/Mu/Beta slopes and amplitudes were estimated in the two experiments is somewhat arbitrary. How were these decided and how robust are the results relative to the choice of these windows? Given the slow nature of the CPP signal (unfolding over a second or more) it would seem that a longer window might be more appropriate to accurately estimate the slope. Relatedly, the choice of a linear line fit needs to be better motivated since the CPP appears to build-up quadratically. Baseline correction might be masking important neural signatures of decision formation in the absence of sensory evidence (i.e. pre-trial influences). Do the main effects presented in the paper persist when CPP, mu/beta and lateralization estimates are derived without any baseline correction?

In light of the reviewers' concern that our choice of amplitude and slope measurement windows may have been arbitrary we have edited the appropriate sections in the methodology of the paper to clearly explain how these windows were chosen based on previously established criteria and adapted to account for certain features of the data that are particular to the present task structure. We have also addressed the reviewers specific concern about whether our findings are robust when using a wider slope measurement window confirming this to be the case. These additions can be found on in the subsection “Analysis of Electrophysiological Signals”.

With regard to baseline correction, we acknowledge that, as the reviewer suggests, there may be interesting pre-trial effects at play within these data. Here we have elucidated one such effect by demonstrating that the timing of evidence accumulation varied from one trial to the next according to the temporal structure of recent trials (see Discussion). However we are precluded from examining other possible pre-trial influences due to the necessity to use baseline correction for broad-band ERP components like the CPP (Luck, 2005). Our choice of baseline window was constrained by certain key factors. Firstly, in Experiment 1 subjects initiated each trial with a simultaneous left/right mouse button click. In addition to preventing us from examining the data before stimulus onset in order to identify pre-trial, cognitive effects, this also prevented us from using a pre-trial window for baseline correction. Secondly, due to the use of a discrete trial task, the onset of the stimulus on each trial gave rise to early visual evoked potentials. In light of these two issues it was necessary to choose a later baseline window to ensure that motor and sensory components had resolved. Our chosen window of 500-550ms appears to precede the onset of the CPP but the possibility cannot be ruled out that some early evidence accumulation was subtracted away. Consequently, we felt it was important also to baseline correct mu/beta to a similar timepoint in order to maximise consistency across these two key signals.

8) Treating the literature: There is a somewhat biased sampling of the literature. There are multiple groups that have provided evidence on the role of CPP in decision making using human electrophysiology and ideally would need to be acknowledged appropriately.

We thank the reviewer for flagging this and we have now included a number of additional references from other lab groups (Introduction, third paragraph). We would be very grateful to the reviewers if they could let us know if we have overlooked any other relevant studies.

9) Data availability: Data sharing is central to eLife's mission as well as the ERC ("FAIR data principles") funding this work. Fully anonymised data from normal control groups as in this study (raw data or subject-wise IVs required to reproduce the findings presented in the main figures) should be shared openly.

We will provide fully anonymised behavioural and neural data in the form of single trial matrices. We would appreciate further guidance on *eLife*’s preferred procedures for sharing large data files (the largest of which is 1.5GB) as we have been unable to upload the files alongside the manuscript submission.

[Editors' note: further revisions were requested prior to acceptance, as described below.]

The manuscript has been improved but there are some remaining issues that need to be addressed before acceptance, as outlined below:[Original comment #3] As reviewer 2 highlights below, the question of how baseline visual or motor activity contributes to biases in evidence accumulation (which can offer mechanistic insights into the role of premature accumulation) has not been fully addressed.We understand that unlike Experiment 2, in Experiment 1, participants made simultaneous left/right button presses to initiate a trial which are likely to contaminate the pre-stimulus period. One possibility to address this issue would be to define a new "baseline" during the presentation of the 50% contrast (ambiguous) stimulus. Since the earliest onset time of the accumulation signal in the current data appears around 800ms after the ambiguous stimulus is presented, this could serve as an objective cutoff for defining a new pre-accumulation baseline window.

We thank the editor and reviewer for these detailed suggestions. Indeed the functional interactions between motor-level and evidence accumulation processes is an important and interesting question. The reviewer’s comments prompted us to more carefully consider the possibility that, in addition to reflecting an accumulation of sensory noise, the pre-evidence build-up of the CPP could also be party driven by baseline variations at the motor-level. In fact, our recent work would lead us to hypothesise that no such relationship should exist. In Steinemann et al., 2018, we showed that while experimental manipulations in speed vs. accuracy emphasis led to substantial adjustments of baseline motor preparation levels, no such adjustments were evident in the CPP. Similarly, in Kelly et al., 2019, predictive cues led to corresponding biases in pre-evidence motor preparation but had no detectable impact on baseline CPP levels. These observations accord with the hypothesis that the CPP provides a relatively ‘pure’ representation of cumulative sensory evidence that is divorced from any strategic adjustments implemented at the motor level. To further test this hypothesis in the present data, we divided the data of each participant into two equal sized bins based on a median split of the single-trial pre-evidence CPP amplitude values (1500-1600ms) and tested for differences in motor-level bias in the preceding time-windows (see Figure 1). This analysis, which is now reported in the subsection “Premature Decision Formation on Trials with Longer Foreperiods” of the revised manuscript indicated that there were no systematic differences in baseline motor bias prior to large versus small pre-evidence CPP build-ups.

[Original comment #6] Please discuss the implications of short-vs.-long foreperiods explicitly, along the lines of what was highlighted in your response to the reviewers. That is, the effect of premature accumulation is likely to manifest only under certain conditions (e.g. when rapid decisions are required).

We thank the reviewer for this suggestion. We have now added additional comment in the Discussion.

[Original comment #9] Data sharing: eLife's list of recommended repositories can be found here: https://fairsharing.org/bsg-p000124/.There are a number of repositories on the list (e.g. OpenNeuro, Dryad etc) that can accommodate the type and size of your data. Please add a link to your shared data in your revised manuscript.

A link can be found to our shared data which we have uploaded to Dryad with user friendly indicator variables included and detailed commentary for anyone who wishes to examine our data (see subsection “EEG acquisition and Preprocessing (Experiment 1 and 2)”.